CERN-TH-2021-130

# Tripartite information at long distances

César A. Agón✶, Pablo Bueno✠ and Horacio Casini✶

✶*Instituto Balseiro, Centro Atómico Bariloche*
*8400-S.C. de Bariloche, Río Negro, Argentina*

✠*CERN, Theoretical Physics Department,*
*CH-1211 Geneva 23, Switzerland*

**Abstract**

We compute the leading term of the tripartite information at long distances for three spheres in a CFT. This falls as $r^{-6\Delta}$, where $r$ is the typical distance between the spheres, and $\Delta$ the lowest primary field dimension. The coefficient turns out to be a combination of terms coming from the two- and three-point functions and depends on the OPE coefficient of the field. We check the result with three-dimensional free scalars in the lattice finding excellent agreement. When the lowest-dimensional field is a scalar, we find that the mutual information can be monogamous only for quite large OPE coefficients, far away from a perturbative regime. When the lowest-dimensional primary is a fermion, we argue that the scaling must always be faster than $r^{-6\Delta_f}$. In particular, lattice calculations suggest a leading scaling $r^{-(6\Delta_f+1)}$. For free fermions in three dimensions, we show that mutual information is also non-monogamous in the long-distance regime.

cesaragon1@gmail.com
pablo.bueno-gomez@cern.ch
casini@cab.cnea.gov.ar

# 1   Introduction

In quantum field theory (QFT), entanglement entropy (EE) characterizes the statistical properties of the vacuum state in the local operator algebras attached to spacetime regions. An important task in investigations related to EE has been to understand how it is related to more traditional QFT observables. Several important connections are well established, such as the realization that renormalization group charges are extractable from the universal parts of the entropy of spheres [1–4].

Universal, cutoff independent pieces of EE can be systematically extracted by considering the

mutual information for two disjoint regions $A, B$,

$$I(A, B) \equiv S(A) + S(B) - S(AB).\qquad(1)$$

This is finite, universal and well defined mathematically. For a conformal field theory (CFT), the renormalization group charges appear in an expansion of the mutual information between two spheres in the short distance limit [4, 5].

In the opposite limit, *i.e.*, for far away regions, application of the replica trick and the operator product expansion (OPE) for twist operators leads to an expansion of mutual information in inverse powers of the distance. The corresponding exponents are sums of the conformal dimensions of the theory [6]. In this way, important information about the spectrum can be recovered from EE. The coefficients in the long-distance expansion can be computed in particular cases. Notably, the exact form of the coefficient of the leading term for spheres has a closed universal expression which only depends on the spin and conformal dimension of the operator [7–9].

In this work we focus on the large separation distances expansion of the tripartite information associated to three disjoint spheres in a CFT. This is defined for three entangling regions $A, B, C$ as

$$\begin{aligned}I_3(A, B, C) &\equiv I(A, B) + I(A, C) - I(A, BC)\qquad(2)\\&= S(A) + S(B) + S(C) - S(AB) - S(AC) - S(BC) + S(ABC).\end{aligned}$$

By its very definition, $I_3$ measures the non-extensivity of mutual information. It is known that $I_3(A, B, C)$ can be used as an order parameter for topological theories [10] and, remarkably, it is always negative for holographic EE [11] —for a discussion on how much tripartite entanglement is present in holographic states see [12] vs [13]. This inequality, $I_3 \leq 0$, called "monogamy" of mutual information,[1] is one of the inequalities defining the so called "holographic entropy cone" [16] —see also [17]. On the other hand, the case $I_3 \equiv 0$ gives place to the "Extensive Mutual Information model" [18], which corresponds to a free fermion in $d = 2$, and has been recently shown not to describe the mutual information of any QFT (or limit of QFTs) in higher dimensions [19]. The case $I_3 \geq 0$ is also known to occur *e.g.,* for free fields [18], so the tripartite does not have a definite sign in general [20, 21].

Part of our interest in the long-distance behavior of $I_3$ arises from the fact that this quantity seems to offer a relatively simple access to the three-point function coefficients —also known as "structure constants" or "OPE coefficients"— which, alongside the conformal dimensions, constitute the CFT data. Here we show that these coefficients already show up in the leading term of the tripartite information. Indeed, when the primary operator with the lowest scaling dimension present in the theory is a scalar, we obtain for three spherical regions $A, B, C$ of radii $R$ and with relative separations $r_{AB}, r_{BC}, r_{AC} \gg R$ ,

$$I_3(A, B, C) = -\frac{R^{6\Delta}}{r_{AB}^{2\Delta} r_{BC}^{2\Delta} r_{AC}^{2\Delta}} \left[ \frac{\sqrt{\pi}}{4} \frac{\Gamma(3\Delta + 1)}{\Gamma\left(3\Delta + \frac{3}{2}\right)} \left(C_{\mathcal{O}\mathcal{O}\mathcal{O}}\right)^2 - \frac{2^{6\Delta}\Gamma\left(\Delta + \frac{1}{2}\right)^3}{2\pi\Gamma\left(3\Delta + \frac{3}{2}\right)} \right],\qquad(3)$$

a formula which is valid in general dimensions. As compared to the analogous expression for the mutual information, a new feature of this expression is its dependence on the structure constant

---

[1] For qubit systems, it was argued in [14] that random states also tend to have a monogamous mutual information. A simple $N$-qubit state which has a positive tripartite information is the GHZ state $1/\sqrt{2}[\otimes_i^N |0\rangle_i + \otimes_i^N |1\rangle_i]$ —see [15] for a discussion on how to construct generalizations of such state which maximize $I_3$.

$C_{\mathcal{O}\mathcal{O}\mathcal{O}}$. This implies that knowledge of the leading term in the tripartite information can be used to extract the values of both the smallest scaling dimension in the theory, $\Delta$, as well as the dynamical coefficient $C_{\mathcal{O}\mathcal{O}\mathcal{O}}$. Thus, considering other primary operator contributions one could imagine extracting as well other OPE coefficients and with this completing the task of getting the full CFT information from the mutual information. When the lowest-dimensional primary is not a scalar, more work is required in order to generalize eq. (3), but we do argue here that the analogous result when this field is a fermion has a vanishing coefficient for the naive leading piece $\sim r^{-6\Delta_f}$.

The remainder of the paper goes as follows. In Section 2 we compute the leading term in the long-distance expansion of the tripartite information for a generic CFT such that its lowest-dimensional primary is a scalar field. In Section 3 we show with an explicit calculation that the term responsible for the would-be leading term in the case of a CFT with a fermionic lowest-dimensional primary identically vanishes. In Section 4 we use lattice calculations in three-dimensions to verify the scalings obtained in the previous sections for free scalars and fermions (in particular, we find a scaling $\sim r^{-(6\Delta_f+1)}$ for the latter). We also verify there that the free scalar result for the three-disks coefficient computed analytically in Section 2 is reproduced numerically in the lattice and we obtain the analogous one for fermions. In Section 5 we conclude with a couple of comments regarding: the implications of our results for the "entropic bootstrap" program; and how difficult is to achieve a monogamous mutual information at long distances. In appendix A we show how our formula for the long-distance tripartite information can be enhanced in order to include the full conformal block associated to the lowest-dimensional primary.

## 2 Tripartite information at long distances

We wish to compute the tripartite information for three entangling regions bounded by spheres of equal radii $R$ in the regime in which the distance between any of the two is much larger than $R$. In order to do this, it is convenient to split $I_3(A, B, C)$ into two contributions, one which depends on the individual mutual informations of pairs of spheres, and a remanent piece which depends only on the subtracted entropy of the three regions, this is

$$I_3(A, B, C) = I(A, B) + I(A, C) + I(B, C) - \tilde{I}_3(A, B, C),\tag{4}$$

where

$$\tilde{I}_3(A, B, C) \equiv S(A) + S(B) + S(C) - S(ABC).\tag{5}$$

We are interested in the leading contribution to $I_3(A, B, C)$ in the long-distance regime of the above set up. For such a computation we can exclusively focus on $\tilde{I}_3(A, B, C)$, since in [7] the corresponding behavior of the remaining mutual informations was already understood.

### 2.1 Warm up: Mutual Information

First, recall that for a given entangling region $A$, the Rényi entropy $S^{(n)}(A)$ can be obtained as the following path integral:

$$S^{(n)}(A) = \frac{1}{1-n} \log \left[ \frac{Z(\mathcal{C}_A^{(n)})}{Z^n} \right],\tag{6}$$

where $\mathcal{C}_A^{(n)}$ represents the replica manifold for the $n$ copies of the original space-time geometry after suitably identifying the region $A$ of copy $i$ with the one of $i+1$, and $n+1 \equiv 1$. $Z(X)$ is the partition function of the theory defined on the manifold $X$ (for simplicity we use $Z$ when the manifold is a single copy of the original spacetime). Using this expression, one gets for the Rényi mutual information

$$I^{(n)}(A,B) = \frac{1}{n-1} \log \left[ \frac{Z(\mathcal{C}_{AB}^{(n)})Z^n}{Z(\mathcal{C}_A^{(n)})Z(\mathcal{C}_B^{(n)})} \right] = \frac{1}{n-1} \log \left[ \frac{Z_{AB}^{(n)} Z^n}{Z_A^{(n)} Z_B^{(n)}} \right], \tag{7}$$

where we have simplified the notation for convenience using $Z_A^{(n)} \equiv Z(\mathcal{C}_A^{(n)})$. In [6], it was proposed that at long distances from the conifold of singularities, one can interpret the associated twist operator as a semi-local operator that couples the $n$ QFT's in the corresponding region. This implies that in the evaluation of the partition function

$$\frac{Z_{AB}^{(n)}}{Z^n} = \langle \Sigma_A^{(n)} \Sigma_B^{(n)} \rangle_{\mathcal{M}^n}, \tag{8}$$

where $\mathcal{M}^n$ is the replicated theory, provided $A$ and $B$ are apart from each other, one can expand $\Sigma_A^{(n)}$ as a linear combination of local operators

$$\Sigma_A^{(n)} = \frac{Z_A^{(n)}}{Z^n} \sum_{\{k_j\}} C_{\{k_j\}}^A \prod_{j=0}^{n-1} \Phi_{k_j}^{(j)}(r_A), \tag{9}$$

where $\{\Phi_{k_j}^{(j)}(r_A)\}$ is a complete set of operators in the $j^{\text{th}}$ copy of the QFT located at a conveniently chosen point $r_A$ in region $A$. We can further separate the identity contributions from the product of operators in (9) as

$$\Sigma_A^{(n)} = \frac{Z_A^{(n)}}{Z^n}(1 + \tilde{\Sigma}_A^{(n)}), \quad \text{where} \quad \tilde{\Sigma}_A^{(n)} = \sum_{\{k_j\} \neq \mathbb{I}} C_{\{k_j\}}^A \prod_{j=0}^{n-1} \Phi_{k_j}^{(j)}(r_A), \tag{10}$$

and analogously for $B$. This leads to

$$\frac{Z_{AB}^{(n)}}{Z^n} = \frac{Z_A^{(n)} Z_B^{(n)}}{Z^{2n}} \left( 1 + \langle \tilde{\Sigma}_A^{(n)} \tilde{\Sigma}_B^{(n)} \rangle_{\mathcal{M}^n} \right), \tag{11}$$

where we take into account that one-point functions vanish in a CFT

$$\langle \tilde{\Sigma}_A^{(n)} \rangle_{\mathcal{M}^n} = 0. \tag{12}$$

The expansion of the logarithm reads

$$\log \left[ \frac{Z_{AB}^{(n)} Z^n}{Z_A^{(n)} Z_B^{(n)}} \right] = \left[ \langle \tilde{\Sigma}_A^{(n)} \tilde{\Sigma}_B^{(n)} \rangle_{\mathcal{M}^n} - \frac{1}{2} \left( \langle \tilde{\Sigma}_A^{(n)} \tilde{\Sigma}_B^{(n)} \rangle_{\mathcal{M}^n} \right)^2 + \cdots \right]. \tag{13}$$

The leading term in the above expansion goes as $(n-1)$ when $n \to 1$, while the higher order terms involve higher powers of $(n-1)$ and as such they vanish in the same limit. The mutual information is thus given entirely by

$$I(A,B) = \lim_{n \to 1} \frac{1}{n-1} \langle \tilde{\Sigma}_A^{(n)} \tilde{\Sigma}_B^{(n)} \rangle_{\mathcal{M}^n}. \tag{14}$$

In [6], it was shown that the leading contribution to $I(A,B)$ comes from products of two operators located at different sheets and, from those, the ones with lowest scaling dimension $\Delta$ contribute the most. Making explicit the contributions of products of the lowest dimensional operator $\mathcal{O}_i$ in different copies, and assuming this operator is a scalar,

$$\tilde{\Sigma}_A^{(n)} = \sum_i C_i^A \mathcal{O}_i + \cdots \sum_{i<j} C_{ij}^A \mathcal{O}^i \mathcal{O}^j + \cdots + \sum_{i<j<k} C_{ijk}^A \mathcal{O}^i \mathcal{O}^j \mathcal{O}^k + \cdots . \tag{15}$$

As $\tilde{\Sigma}_A^{(n)}$ vanish in the $n \to 0$ limit (where $\Sigma_A^1 = 1$) the coefficient of the linear term must be proportional to $n - 1$ and will not contribute to the mutual information [7]. Then the leading contribution in the long-distance expansion has the form

$$I(A,B) = \left( \lim_{n \to 1} \frac{1}{n-1} \sum_{i<j} C_{ij}^A C_{ij}^B \right) \frac{1}{r^{4\Delta}} + \cdots \tag{16}$$

The coefficients $C_{ij}^A$ are given by the two-point functions on the conifold properly normalized [6], this is

$$C_{ij}^A = \lim_{r \to \infty} |r|^{4\Delta} \langle \mathcal{O}^i(r) \mathcal{O}^j(r) \rangle_{\mathcal{C}_A^{(n)}} . \tag{17}$$

Although it might be difficult to have an analytic handle on the above coefficients, the factor in brackets appearing in (16) can actually be evaluated analytically [7]. The result is

$$\lim_{n \to 1} \frac{1}{n-1} \sum_{i<j} C_{ij}^A C_{ij}^B = \frac{\sqrt{\pi}}{4} \frac{\Gamma(2\Delta + 1)}{\Gamma\left(2\Delta + \frac{3}{2}\right)} R_A^{2\Delta} R_B^{2\Delta} . \tag{18}$$

Taking $R_A = R_B = R$ for simplicity, we can write the leading term in the mutual information as

$$I(A,B) = \frac{\sqrt{\pi}}{4} \frac{\Gamma(2\Delta + 1)}{\Gamma\left(2\Delta + \frac{3}{2}\right)} \frac{R^{4\Delta}}{r^{4\Delta}} + \cdots . \tag{19}$$

The next term in the expansion of the twist operator which contributes to $I(A,B)$ (assuming there are no other operators with dimension $\Delta_\phi \leq 3\Delta/2$) is

$$\sum_{i<j<k} C_{ijk}^A \mathcal{O}^i \mathcal{O}^j \mathcal{O}^k . \tag{20}$$

As we will argue later, such type of terms would give a contribution to $I(A,B)$ of order $\sim (R/r)^{6\Delta}$. Indeed, this type of contributions were computed in [22], for free scalars in three dimensions. We show below that this order of contribution appears also in $\tilde{I}_3(A,B,C)$ and that it is in fact the leading one in $I_3(A,B,C)$.

## 2.2 Tripartite Information

Let us now move to the tripartite information. We consider three well separated spheres and compute the leading term, assuming the lowest-dimensional operator is a scalar. For the evaluation

of both $\tilde{I}_3(A, B, C)$ and $I_3(A, B, C)$, the new ingredient is the computation of $Z_{ABC}^{(n)}$. This can be expressed as

$$\frac{Z_{ABC}^{(n)}}{Z^n} = \langle \Sigma_A^{(n)} \Sigma_B^{(n)} \Sigma_C^{(n)} \rangle_{\mathcal{M}^n}, \tag{21}$$

which in terms of correlators of $\tilde{\Sigma}$'s results in

$$\begin{aligned}
\frac{Z_{ABC}^{(n)}}{Z^n} &= \frac{Z_A^{(n)} Z_B^{(n)} Z_C^{(n)}}{Z^{3n}} \times \\
&+ \left( 1 + \langle \tilde{\Sigma}_B^{(n)} \tilde{\Sigma}_C^{(n)} \rangle_{\mathcal{M}^n} + \langle \tilde{\Sigma}_A^{(n)} \tilde{\Sigma}_C^{(n)} \rangle_{\mathcal{M}^n} + \langle \tilde{\Sigma}_A^{(n)} \tilde{\Sigma}_B^{(n)} \rangle_{\mathcal{M}^n} + \langle \tilde{\Sigma}_A^{(n)} \tilde{\Sigma}_B^{(n)} \tilde{\Sigma}_C^{(n)} \rangle_{\mathcal{M}^n} \right),
\end{aligned} \tag{22}$$

where once again we eliminated terms with a single $\tilde{\Sigma}$ as they have zero expectation value. This expansion implies the following leading contribution to $\tilde{I}_3(A, B, C)$,

$$\begin{aligned}
\tilde{I}_3^{(n)}(A, B, C) &= \frac{1}{n-1} \log \left[ \frac{Z_{ABC}^{(n)} Z^{2n}}{Z_A^{(n)} Z_B^{(n)} Z_C^{(n)}} \right] = \frac{1}{n-1} \log \left[ 1 + \left( \frac{Z_{ABC}^{(n)} Z^{2n}}{Z_A^{(n)} Z_B^{(n)} Z_C^{(n)}} - 1 \right) \right] \\
&= \frac{1}{n-1} \left[ \left( \frac{Z_{ABC}^{(n)} Z^{2n}}{Z_A^{(n)} Z_B^{(n)} Z_C^{(n)}} - 1 \right) - \frac{1}{2} \left( \frac{Z_{ABC}^{(n)} Z^{2n}}{Z_A^{(n)} Z_B^{(n)} Z_C^{(n)}} - 1 \right)^2 + \cdots \right].
\end{aligned} \tag{23}$$

The linear term in the expansion of the logarithm goes as $\sim (n-1)$ in the $n \to 1$ limit, while the other terms have higher powers. Therefore, for the purpose of computing the tripartite information only the first term contributes, and we get

$$\begin{aligned}
\tilde{I}_3(A, B, C) &= \lim_{n \to 1} \frac{1}{n-1} \left[ \langle \tilde{\Sigma}_B^{(n)} \tilde{\Sigma}_C^{(n)} \rangle_{\mathcal{M}^n} + \langle \tilde{\Sigma}_A^{(n)} \tilde{\Sigma}_C^{(n)} \rangle_{\mathcal{M}^n} \right. \\
&\qquad \left. + \langle \tilde{\Sigma}_A^{(n)} \tilde{\Sigma}_B^{(n)} \rangle_{\mathcal{M}^n} + \langle \tilde{\Sigma}_A^{(n)} \tilde{\Sigma}_B^{(n)} \tilde{\Sigma}_C^{(n)} \rangle_{\mathcal{M}^n} \right].
\end{aligned} \tag{24}$$

In the above equation we can recognize the leading-order expressions (in powers of $(n-1)$) for the Rényi mutual informations of pairs of regions (13). After such identification, we can rewrite (24) as

$$\tilde{I}_3(A, B, C) = \lim_{n \to 1} \frac{1}{n-1} \langle \tilde{\Sigma}_A^{(n)} \tilde{\Sigma}_B^{(n)} \tilde{\Sigma}_C^{(n)} \rangle_{\mathcal{M}^n} + I(A, B) + I(B, C) + I(A, C). \tag{25}$$

Comparing this equation with (4) we straightforwardly identify an exact expression for the tripartite information,

$$I_3(A, B, C) = \lim_{n \to 1} \frac{1}{1-n} \langle \tilde{\Sigma}_A^{(n)} \tilde{\Sigma}_B^{(n)} \tilde{\Sigma}_C^{(n)} \rangle_{\mathcal{M}^n}. \tag{26}$$

The lowest order of approximation corresponds to taking the quadratic term in the expansion (15). Thus

$$\begin{aligned}
&\langle \tilde{\Sigma}_A^{(n)} \tilde{\Sigma}_B^{(n)} \tilde{\Sigma}_C^{(n)} \rangle_{\mathcal{M}^n} \\
&= \sum_{\{k_j\}} \sum_{\{p_l\}} \sum_{\{q_m\}} C_{\{k_j\}}^A C_{\{p_l\}}^B C_{\{q_m\}}^C \prod_{j,l,m=0}^{n-1} \langle \Phi_{k_j}^{(j)}(r_A) \Phi_{p_l}^{(l)}(r_B) \Phi_{q_m}^{(m)}(r_C) \rangle_{\mathcal{M}^n} \\
&\sim \sum_{ij} \sum_{kl} \sum_{pq} C_{ij} C_{kl} C_{pq} \langle \mathcal{O}^i(r_A) \mathcal{O}^j(r_A) \mathcal{O}^k(r_B) \mathcal{O}^l(r_B) \mathcal{O}^p(r_C) \mathcal{O}^q(r_C) \rangle_{\mathcal{M}^n}.
\end{aligned} \tag{27}$$

Within the correlators, we need to pair operators of different regions. They will only give non-zero contributions provided they are in the same sheet. We can describe two different configurations that contribute in these sums. In order to analyze them, it is convenient to introduce a matrix representation. Since we have $n$ sheets and 3 regions, we can put the various operator locations in a $3 \times n$ matrix as follows,

$$\text{First configuration:} \quad \begin{pmatrix} 1 & \cdots \mathcal{O}^i(r_A) & \cdots \mathcal{O}^j(r_A) & \cdots 1 \\ 1 & \cdots \mathcal{O}^i(r_B) & \cdots \mathcal{O}^j(r_B) & \cdots 1 \\ 1 & \cdots \mathcal{O}^i(r_C) & \cdots \mathcal{O}^j(r_C) & \cdots 1 \end{pmatrix}, \tag{28}$$

$$\text{Second configuration:} \quad \begin{pmatrix} 1 & \cdots \mathcal{O}^i(r_A) & \cdots \mathcal{O}^j(r_A) & \cdots 1 & \cdots 1 \\ 1 & \cdots 1 & \cdots \mathcal{O}^j(r_B) & \cdots \mathcal{O}^k(r_B) & \cdots 1 \\ 1 & \cdots \mathcal{O}^i(r_C) & \cdots 1 & \cdots \mathcal{O}^k(r_C) & \cdots 1 \end{pmatrix}. \tag{29}$$

Here, each row represents the operators associated to a given region: $A$, $B$, $C$ and each column represents a sheet on the multiple copies of the geometry.

We normalize primary operators so that their two- and three-point functions are given by

$$\langle \mathcal{O}(r_A)\mathcal{O}(r_B) \rangle = \frac{1}{r_{AB}^{2\Delta}}, \qquad \text{and} \qquad \langle \mathcal{O}(r_A)\mathcal{O}(r_B)\mathcal{O}(r_C) \rangle = \frac{C_{\mathcal{O}\mathcal{O}\mathcal{O}}}{r_{AB}^{\Delta} r_{BC}^{\Delta} r_{AC}^{\Delta}}, \tag{30}$$

respectively. We use the notation $r_{AB} = |r_A - r_B|$. In the first configuration we get a product of two three-point functions while in the second we get a product of three two-point functions. Both of them yield a $\sim (r_{AB} r_{BC} r_{AC})^{-2\Delta}$ behavior.

A configuration of the type (28) for fixed $\{i, j\}$ is unique while a configuration like (29) for fixed $\{i, j, k\}$ —via permutations across the regions $A$, $B$ and $C$— gives rise to $3! = 6$ non-equivalent ones but with the same numerical value. Thus, the full answer is given by

$$I_3(A, B, C) = -\frac{1}{r_{AB}^{2\Delta} r_{BC}^{2\Delta} r_{AC}^{2\Delta}} \lim_{n \to 1} \left[ \frac{(C_{\mathcal{O}\mathcal{O}\mathcal{O}})^2}{n-1} \sum_{i<j} (C_{ij})^3 + \frac{6}{n-1} \sum_{i<j<k} C_{ij} C_{jk} C_{ki} \right]. \tag{31}$$

By looking at the derivation of (18) in [7], we observe that for a power different than two, say $s$, we simply need to replace $\Delta \to s\Delta/2$ in that formula. For our case, $s = 3$, and we obtain

$$\lim_{n \to 1} \frac{1}{n-1} \sum_{i<j} (C_{ij})^3 = \frac{\sqrt{\pi}}{4} \frac{\Gamma(3\Delta + 1)}{\Gamma\left(3\Delta + \frac{3}{2}\right)} R_A^{2\Delta} R_B^{2\Delta} R_C^{2\Delta}. \tag{32}$$

This allows us to evaluate the first term of (31), which therefore gives a negative contribution to the tripartite information.

The second term is more complicated to analyze. We devote section 2.3 to explain how to compute it using the same techniques introduced in [7]. The final result reads

$$\lim_{n \to 1} \frac{1}{n-1} \sum_{i<j<k} C_{ij} C_{jk} C_{ki} = -\frac{2^{6\Delta} \Gamma\left(\Delta + \frac{1}{2}\right)^3}{12\pi \Gamma\left(3\Delta + \frac{3}{2}\right)} R_A^{2\Delta} R_B^{2\Delta} R_C^{2\Delta}. \tag{33}$$

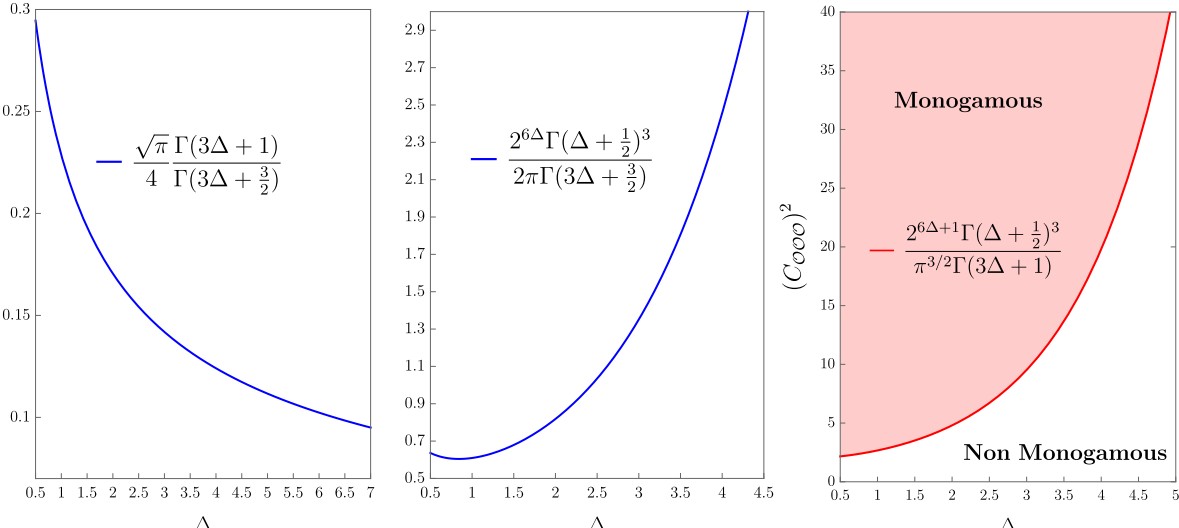

Figure 1: In the first two plots, we show, respectively, the dependence on the lowest-scaling dimension $\Delta$ of a given CFT of the two coefficients appearing in our formula for the tripartite information at long distances. The third plot represents how the value of $(C_{\mathcal{O}\mathcal{O}\mathcal{O}})^2$ determines whether or not the CFT has a monogamous mutual information (at least in the long-distance regime). When $(C_{\mathcal{O}\mathcal{O}\mathcal{O}})^2$ is larger than $\frac{2^{6\Delta+1}\Gamma(\Delta+1/2)^3}{\pi^{3/2}\Gamma(3\Delta+1)}$ (red curve) the mutual information is monogamous and viceversa.

Putting the pieces together, we obtain a closed expression for the leading term in the long-distance regime of the tripartite information, namely,

$$I_3(A,B,C) = -\frac{R_A^{2\Delta}R_B^{2\Delta}R_C^{2\Delta}}{r_{AB}^{2\Delta}r_{BC}^{2\Delta}r_{AC}^{2\Delta}}\left[\frac{\sqrt{\pi}}{4}\frac{\Gamma(3\Delta+1)}{\Gamma\left(3\Delta+\frac{3}{2}\right)}\left(C_{\mathcal{O}\mathcal{O}\mathcal{O}}\right)^2 - \frac{2^{6\Delta}\Gamma\left(\Delta+\frac{1}{2}\right)^3}{2\pi\Gamma\left(3\Delta+\frac{3}{2}\right)}\right]. \tag{34}$$

This is our main result. Observe that both terms inside the square brackets are positive-definitive except for the relative minus sign. As it happens for the long distance coefficient of the mutual information, the coefficient of the tripartite information depends on the lowest scaling dimension but not explicitly on the spacetime dimension. This is due to the universal form of the modular flow for spherical entangling surfaces. The coefficient in front of the $(C_{\mathcal{O}\mathcal{O}\mathcal{O}})^2$ is a monotonically decreasing function of $\Delta$ and tends to zero for $\Delta \gg 1$. On the other hand, the coefficient with the minus sign takes a minimum value of $\simeq 0.604$ for $\Delta_{\min} \simeq 0.841$ and then becomes monotonically increasing for greater values of $\Delta$ —see left and middle plots in Fig. 1. We observe then that depending on the value of $C_{\mathcal{O}\mathcal{O}\mathcal{O}}$, the tripartite information in this regime can be positive, negative or zero. The first two cases correspond to non-monogamous and monogamous mutual informations, respectively —see right plot in Fig. 1. We make more comments regarding these possibilities in Section 5.

Note also that when the coefficient $C_{\mathcal{O}\mathcal{O}\mathcal{O}} = 0$ —in particular when the lowest dimensional operator is free or charged under a global symmetry that gives non-zero charge to the product of three operators (such as a $\mathbb{Z}_2$ symmetry acting as $\mathcal{O} \to -\mathcal{O}$)— the tripartite information reduces to

$$I_3(A,B,C) = \frac{2^{6\Delta}\Gamma\left(\Delta+\frac{1}{2}\right)^3}{2\pi\Gamma\left(3\Delta+\frac{3}{2}\right)}\frac{R_A^{2\Delta}R_B^{2\Delta}R_C^{2\Delta}}{r_{AB}^{2\Delta}r_{BC}^{2\Delta}r_{AC}^{2\Delta}}. \tag{35}$$

For all these theories, the mutual information is non monogamous. In particular for a free scalar in $d$ spacetime dimensions, $\Delta = (d-2)/2$, and setting the three radii of the spheres equal for simplicity, one gets,

$$I_3(A, B, C)|_{d=3}^{\text{scalar}} = \frac{2^{(3d-7)}\Gamma\left(\frac{d-1}{2}\right)^3}{\pi\Gamma\left(\frac{3(d-1)}{2}\right)} \frac{R^{3(d-2)}}{r_{AB}^{(d-2)} r_{BC}^{(d-2)} r_{AC}^{(d-2)}} \,. \tag{36}$$

Later, we will verify this expression in the lattice for $d = 3$. In that case, we have

$$I_3(A, B, C)|_{d=3}^{\text{scalar}} = \frac{2}{\pi} \frac{R^3}{r_{AB} r_{BC} r_{AC}} \,. \tag{37}$$

Another natural example is the Ising model in three dimensions. The lowest scaling dimension is in that case given by [23] $\Delta_{d=3}^{\text{Ising}} = 0.5181489(10)$, and hence we find for the corresponding long-distance tripartite information

$$I_3(A, B, C)|_{d=3}^{\text{Ising}} \simeq 0.632833 \frac{R^{3.10889}}{r_{AB}^{1.0362978} r_{BC}^{1.0362978} r_{AC}^{1.0362978}} \,. \tag{38}$$

Note that the power is slightly greater than in the scalar case, whereas the coefficient is smaller $(2/\pi \simeq 0.63662)$. Similarly, for the $O(2)$ model one finds using results from [24],

$$I_3(A, B, C)|_{d=3}^{O(2)} \simeq 0.632645 \frac{R^{3.11453}}{r_{AB}^{1.03818} r_{BC}^{1.03818} r_{AC}^{1.03818}} \,. \tag{39}$$

For the $O(3)$ model, the currently known result for $\Delta_{d=3}^{O(3)}$ [23] suggests that it may be slightly greater than $\Delta_{d=3}^{O(2)}$, which would produce a greater power for $R$ and $r$, and a slightly smaller coefficient. For sufficiently large values of $N$, the result tends to the free scalar values. In particular, in the large-$N$ limit, we have

$$I_3(A, B, C)|_{d=3}^{O(N\gg1)} \simeq \left[\frac{2}{\pi} + \frac{4(4\log(2) - 3)}{\pi^3 N}\right] \frac{R^{3+8/(\pi^2 N)}}{r_{AB}^{1+8/(3\pi^2 N)} r_{BC}^{1+8/(3\pi^2 N)} r_{AC}^{1+8/(3\pi^2 N)}} \,, \tag{40}$$

where we used the expression for $\Delta|_{d=3}^{O(N\gg1)}$ valid up to $\mathcal{O}(1/N)$ —see *e.g.,* [25] for the answer up to $\mathcal{O}(1/N^3)$.

Our formula is completely general, so it applies to any other model with a scalar as its lowest scaling dimension operator. For instance, the explicit expression for the $O(N)$ model in the large-$N$ expansion for general $d$ can be similarly obtained using *e.g.,* results from [26]. Our formula can also be generalized to include all the descending operators associated to the leading term in the OPE expansion of the twist operator, this is, the quadratic term in (15). The final formula is given in (119). We discussed this generalization in detail in appendix A.

## 2.3 Analytic continuation of the sums over coefficients $C_{ij}$

The first coefficient in our formula eq. (34) can be relatively easily obtained, as we saw in the previous subsection. On the other hand, computing the second one has required considerably more

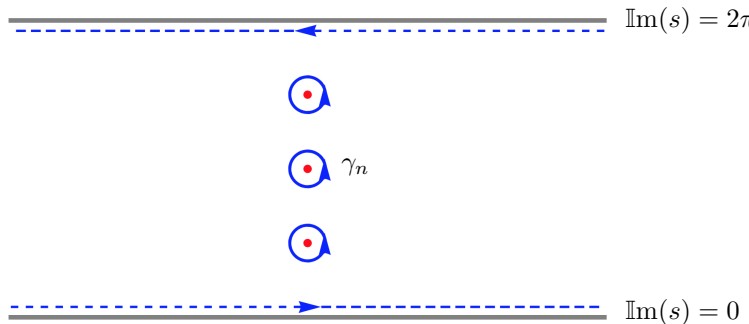

Figure 2: Integration contour (depicted in solid blue) used to evaluate the complex integral in (43). Assuming the integrand vanishes for $\mathbb{Re}(s) \to \pm\infty$, we can then deform the contour $\gamma_n$ to be the dashed blue lines at $\mathbb{Im}(s) = \epsilon$ and at $\mathbb{Im}(s) = 2\pi n - \epsilon$ with $\epsilon > 0$. For illustrative purposes, we have picked the value $n = 4$ to make this figure.

work, which we present here. We wish to show that the LHS of eq. (33) can be written as the expression appearing in the RHS.

As a first step, we recall that in [7], the coefficient $C_{ij}$ was related to the thermal Green function of the theory on hyperbolic space, evaluated at different points along the thermal circle

$$C_{jj'} = (2R)^{2\Delta} G_n(2\pi(j - j')) \,, \tag{41}$$

where the factor $(2R)^{2\Delta}$ comes from the details of the conformal transformation. More explicitly, the conformal map introduced in [4] takes a single copy of $\mathbb{R}^d$ into $\mathbb{S}^1 \times \mathbb{H}^{d-1}$, where $\mathbb{H}^{d-1}$ is the hyperbolic space. Such map can be adapted such that the conifold of singularities $\mathcal{C}_A^{(n)}$ is mapped to $\mathbb{S}_n^1 \times \mathbb{H}^{d-1}$, where the thermal circle $\mathbb{S}_n^1$ now obeys $\tau \equiv \tau + 2\pi n$ and thus allows us to connect two-point functions on $\mathcal{C}_A^{(n)}$ with thermal two-point functions on $\mathbb{S}_n^1 \times \mathbb{H}^{d-1}$ (41).

For $n = 1$ the thermal two-point function is known to be

$$G_1(\tau) = \frac{1}{2^\Delta (1 - \cos\tau)^\Delta} = \frac{1}{2^{2\Delta} \sin^{2\Delta}(\tau/2)} \,, \tag{42}$$

where we also known that $G_1(-is)$ decays as $e^{-\Delta|s|}$ for real $s$. The assumed analyticity in $n$ implies that a similar exponential decay should happen for $G_n(-is)$. This property together with the standard analyticity properties of thermal two-point functions allows us to evaluate the sum $\sum_{j=1}^{n-1} G_n^2(2\pi j)$ in [7]. A key step in that computation is to relate the previous sum to a contour integral

$$\sum_{j=1}^{n-1} G_n^2(2\pi j) = \int_{\gamma_n} \frac{\mathrm{d}s}{2\pi i} \frac{G_n^2(-is)}{e^s - 1} \,, \tag{43}$$

where the exponential decay assumption allows us to deform the integral contour $\gamma_n$ to the horizontal lines at $\mathbb{Im}(s) = 2\pi n - \epsilon$ and $\mathbb{Im}(s) = \epsilon$ as depicted in Figure 2.

Here we are interested in the following sum,

$$\sum_{i<j<k} C_{ij} C_{jk} C_{ki} = (2R)^{6\Delta} \sum_{i<j<k} G_n(2\pi(j - i)) G_n(2\pi(k - j)) G_n(2\pi(i - k)) \,, \tag{44}$$

where we used eq. (41) in the RHS. Notice that one can extend the sum to a disordered one and pay a symmetry factor for it. Then one can further fix the location of one of the operators to be zero and multiply by $n$ using the replica symmetry. After that, we can recover an ordered sum by paying the price of a remaining symmetry factor of 2. The sequence is

$$\sum_{i<j<k} \to \frac{1}{3!} \sum_{i\neq j\neq k} \to \frac{n}{3!} \sum_{i=0,j\neq k>0} \to \frac{n}{3} \sum_{i=0,k>j>0} . \tag{45}$$

Applying the above equivalence we can write

$$\sum_{i<j<k} C_{ij}C_{jk}C_{ki} = 2^{6\Delta} R^{6\Delta} \frac{n}{3} C_n , \tag{46}$$

where

$$C_n \equiv \sum_{j=1}^{n-2} G_n(2\pi j) \sum_{k=j+1}^{n-1} G_n(2\pi(k-j))G_n(2\pi k) , \tag{47}$$

and where we also used the fact that Green functions must be reflection symmetric $G_n(\tau) = G_n(-\tau)$. Now it is convenient to rewrite the double sum as

$$C_n = \sum_{k=2}^{n-1} G_n(2\pi k) \sum_{j=1}^{k-1} G_n(2\pi(j-k))G_n(2\pi j) , \tag{48}$$

where the relevant contour for the $j$-sum is given in Fig. 2 (with the simple $n \to k$ replacement). At this point we cannot make a similar replacement for the other sum as the first contour depends on the integer label $k$. We deform the integral to the horizontal contours along $\mathbb{Im}(s) = \epsilon$ and $\mathbb{Im}(s) = 2\pi k - \epsilon$. The vertical parts do not contribute as we assume an exponential decay along the imaginary axis as discussed around (42). Then, we have

$$C_n = \sum_{k=2}^{n-1} G_n(2\pi k) \int_{-\infty}^{\infty} \frac{ds}{2\pi i} \left[ \frac{G_n(-is+\epsilon)G_n(-is-2\pi k+\epsilon)}{e^{(s+i\epsilon)}-1} \right. \tag{49}$$
$$\left. - \frac{G_n(-is-\epsilon)G_n(-is+2\pi k-\epsilon)}{e^{(s+i2\pi k-i\epsilon)}-1} \right] .$$

Using $e^{2\pi i k} = 1$ for integer $k$ one gets

$$C_n = \sum_{k=2}^{n-1} G_n(2\pi k) \int_{-\infty}^{\infty} \frac{ds}{2\pi i} \left[ \frac{G_n(-is+\epsilon)G_n(-is-2\pi k+\epsilon)}{e^{(s+i\epsilon)}-1} \right. \tag{50}$$
$$\left. - \frac{G_n(-is-\epsilon)G_n(-is+2\pi k-\epsilon)}{e^{(s-i\epsilon)}-1} \right] .$$

The remanent sum can be done via a contour integral by introducing the following function [9]

$$f(n,\tau) \equiv \frac{1}{2\pi i} \sum_{k=2}^{n-1} \frac{1}{\tau-ik} = \frac{1}{2\pi} \left( \psi(n+i\tau) - \psi(2+i\tau) \right) \tag{51}$$

where $\psi(z) \equiv \Gamma'(z)/\Gamma(z)$ is the digamma function. For positive integer $n \geq 3$, the function $f(n, iu)$ has poles at $u = 2, \cdots, n-1$ with residue one, thus, one can turn the sum over $k$ in (50) into a contour integral over $\tau$ with with $k \to -i\tau$ as

$$C_n = \int_{-\infty}^{\infty} \frac{\mathrm{d}s}{2\pi i} \oint \mathrm{d}\tau f(n, \tau) \, G_n(-2\pi i\tau) \left[ \frac{G_n(-is + \epsilon)G_n(-is + 2\pi i\tau + \epsilon)}{e^{(s+i\epsilon)} - 1} \right. \tag{52}$$
$$\left. - \frac{G_n(-is - \epsilon)G_n(-is - 2\pi i\tau - \epsilon)}{e^{(s-i\epsilon)} - 1} \right] .$$

The above contour must encircle the poles along $\tau = iu$ with $u \geq 2$. Fixing the integration contour one can study the $n \to 1$ limit of the above expression. Before doing so let us study in some detail the function $f(n, \tau)$. First, notice that one can rewrite (51) as

$$f(n, \tau) = -\frac{1}{2\pi i} \frac{1}{\tau - i} + \frac{1}{2\pi} \left( \psi(n + i\tau) - \psi(1 + i\tau) \right) , \tag{53}$$

where we have added un subtracted a function with a single pole at $\tau = i$ and used the following recursive property $\psi(z + 1) = \psi(z) + 1/z$. Of course, the full function has no poles at $\tau = i$, nevertheless, such separation is convenient since the second term has a simple expression in the $n \to 1$ limit [9]. Since we want to discard the first term, it is enough to make sure the integration contour in (52) does not contain the spurious pole at $\tau = i$. Indeed, we will chose the contour integral to be made out of the line $\mathrm{Im}(\tau) = 3i/2$ plus a semi-circle of infinite radius closing the contour on the upper half plane. Such contour satisfies all our requirements and the integral on the semi-circle vanishes due to the exponential damp coming from $G_n$.

In the $n \to 1$ limit we have

$$f(n, \tau) \sim -\frac{1}{2\pi i} \frac{1}{\tau - i} - (n-1) \left( \frac{1}{2\pi} \psi'(-i\tau) + \frac{\pi}{2\sinh^2(\pi\tau)} \right) + \mathcal{O}((n-1)^2) . \tag{54}$$

The function $\psi'(-i\tau)$ and $1/(\tau - i)$ above has no poles inside the integration contour and therefore, they do not contribute to the contour integral. The second term inside the parenthesis gives a contribution proportional to $n - 1$, which means we can evaluate the remaining terms in the integrand at $n = 1$. Thus, the leading term in the $n - 1$ expansion of $C_n$ is

$$C_n = (n-1) \int_{-\infty}^{\infty} \frac{\mathrm{d}s}{2\pi i} \int_{\infty}^{\infty} \mathrm{d}s' \frac{G_1(-is' + \pi)}{4\cosh^2(s'/2)} \left[ \frac{G_1(-is + \epsilon)G_1(-i(s - s') + \pi + \epsilon)}{e^{(s+i\epsilon)} - 1} \right. \tag{55}$$
$$\left. - \frac{G_1(-is - \epsilon)G_1(-i(s + s') + \pi - \epsilon)}{e^{(s-i\epsilon)} - 1} \right] ,$$

where we have changed the integration variable from $\tau$ to $s'$ via $\tau = 3i/2 + s'/2\pi$, and used the $2\pi$ periodicity of $G_1(\tau)$. The integral above can be further simplified into

$$C_n = (n-1) \int_{-\infty}^{\infty} \frac{\mathrm{d}s}{2\pi i} \left[ \frac{G_1(-is + \epsilon)}{e^{(s+i\epsilon)} - 1} - \frac{G_1(-is - \epsilon)}{e^{(s-i\epsilon)} - 1} \right] \int_{-\infty}^{\infty} \mathrm{d}s' \frac{G_1(-is' + \pi)G_1(-i(s - s') + \pi)}{4\cosh^2(s'/2)} , \tag{56}$$

which we arrive at after changing $s' \to -s'$ in the second integral, using the reflection symmetry of $G_1(\tau)$ and the $2\pi$ periodicity. We also dropped the $\epsilon$ dependence on the $G_1$ functions with real

argument $\pi$ as those functions are completely regular inside the integrals. Replacing the expression for $G_1$ in the second integral leads to

$$C_n = (n-1)\frac{1}{2^{4\Delta}}\int_{-\infty}^{\infty}\frac{ds}{2\pi i}\left[\frac{G_1(-is+\epsilon)}{e^{(s+i\epsilon)}-1}-\frac{G_1(-is-\epsilon)}{e^{(s-i\epsilon)}-1}\right] \tag{57}$$
$$\cdot\int_{-\infty}^{\infty}ds'\frac{1}{4\cosh^{2(\Delta+1)}(s'/2)\cosh^{2\Delta}((s-s')/2)} \; .$$

So far we have succeeded at obtaining a closed-form expression for the linear piece of $C_n$ in the $(n-1)$ expansion, which is relevant for the computation of the tripartite information. This expression is given as a double integral which we will now evaluate via a series of convenient manipulations. Let us first separate $C_n$ into two contributions as

$$C_n = C_n^+ - C_n^- \; , \tag{58}$$

with the obvious identifications. Now, let us factor out the coupling term in the double integral by introducing a delta function of the form

$$\int_{-\infty}^{\infty}\frac{du}{2}\delta\left[\frac{u}{2}-\left(\frac{s-s'}{2}\right)\right]=\int_{-\infty}^{\infty}\frac{dq}{2\pi}\int_{-\infty}^{\infty}\frac{du}{2}e^{iq\left[\frac{u}{2}-\left(\frac{s-s'}{2}\right)\right]} \; . \tag{59}$$

Then, the $C_n^{\pm}$ become

$$C_n^{\pm} = (n-1)\frac{1}{2^{4\Delta+3}}\int_{-\infty}^{\infty}\frac{dq}{2\pi}\int_{-\infty}^{\infty}\frac{ds}{2\pi i}\frac{e^{-iqs/2}G_1(-is\pm\epsilon)}{e^{(s\pm i\epsilon)}-1}\int_{-\infty}^{\infty}ds'\frac{e^{iqs'/2}}{\cosh^{2(\Delta+1)}(s'/2)}$$
$$\int_{-\infty}^{\infty}du\frac{e^{iqu/2}}{\cosh^{2\Delta}(u/2)} \; . \tag{60}$$

Now, the $s$ integral in this $C_n^{\pm}$ contour can be deformed to the $\mathrm{Im}(s) = \pm i\pi$ surface and after that we can safely take $\epsilon \to 0$. This results in

$$C_n^{\pm} = -(n-1)\frac{1}{2^{6\Delta+4}}\int_{-\infty}^{\infty}\frac{dq}{2\pi}e^{\pm q\pi/2}\int_{-\infty}^{\infty}\frac{ds}{2\pi i}\frac{e^{-s/2}e^{-iqs/2}}{\cosh^{2\Delta+1}\left(\frac{s}{2}\right)}\int_{-\infty}^{\infty}ds'\frac{e^{iqs'/2}}{\cosh^{2(\Delta+1)}(s'/2)}$$
$$\int_{-\infty}^{\infty}du\frac{e^{iqu/2}}{\cosh^{2\Delta}(u/2)} \; , \tag{61}$$

and therefore for $C_n$ we get

$$C_n = -(n-1)\frac{1}{2^{6\Delta+3}}\frac{1}{2\pi i}\int_{-\infty}^{\infty}\frac{dq}{2\pi}\sinh\left(\frac{q\pi}{2}\right)\int_{-\infty}^{\infty}ds\frac{e^{-s/2}e^{-iqs/2}}{\cosh^{2\Delta+1}\left(\frac{s}{2}\right)}\int_{-\infty}^{\infty}ds'\frac{e^{iqs'/2}}{\cosh^{2(\Delta+1)}(s'/2)}$$
$$\int_{-\infty}^{\infty}du\frac{e^{iqu/2}}{\cosh^{2\Delta}(u/2)} \; . \tag{62}$$

Now we can use the following integral

$$\int_{-\infty}^{\infty}du\frac{e^{\pm iqu/2}}{\cosh^{\Delta}(u/2)}=\frac{2^{\Delta}\Gamma\left(\frac{\Delta}{2}+i\frac{q}{2}\right)\Gamma\left(\frac{\Delta}{2}-i\frac{q}{2}\right)}{\Gamma(\Delta)} \; , \tag{63}$$

which can be analytically continued to get

$$\int_{-\infty}^{\infty}du\frac{e^{\pm iqu/2}e^{-u/2}}{\cosh^{\Delta}(u/2)}=\frac{2^{\Delta}\Gamma\left(\frac{\Delta}{2}\pm i\frac{q}{2}-\frac{1}{2}\right)\Gamma\left(\frac{\Delta}{2}\mp i\frac{q}{2}+\frac{1}{2}\right)}{\Gamma(\Delta)} \; . \tag{64}$$

Replacing these integrals in the resulting expression for $C_n$ one gets:

$$C_n = -\frac{(n-1)}{2\pi i} \int_{-\infty}^{\infty} \frac{dq}{2\pi} \sinh\left(\frac{q\pi}{2}\right) \frac{\Gamma\left(\Delta - i\frac{q}{2}\right)^2 \Gamma\left(\Delta + i\frac{q}{2} + 1\right)^2 \Gamma\left(\Delta - i\frac{q}{2} + 1\right) \Gamma\left(\Delta + i\frac{q}{2}\right)}{\Gamma(2\Delta + 1)\Gamma(2\Delta + 2)\Gamma(2\Delta)}.$$
(65)

This expression is not obviously real. However, we can rewrite it in a manifestly real form using the relation $|\Gamma(z)|^2 = \Gamma(z)\Gamma(\bar{z})$ and the defining property of the Gamma function as

$$C_n = -\frac{(n-1)}{2\pi i} \int_{-\infty}^{\infty} \frac{dq}{2\pi} \sinh\left(\frac{q\pi}{2}\right) \left(\Delta + i\frac{q}{2}\right) \frac{\left|\Gamma\left(\Delta + i\frac{q}{2} + 1\right)\right|^2 \left|\Gamma\left(\Delta + i\frac{q}{2}\right)\right|^4}{\Gamma(2\Delta + 1)\Gamma(2\Delta + 2)\Gamma(2\Delta)}.$$
(66)

In this integral, only the even part contributes. Since $|\Gamma(z)|^2$ is even on the imaginary part of its argument we conclude that only the imaginary part in $(\Delta + iq/2)$ contributes. This results in[2]

$$C_n = -(n-1)\frac{1}{8\pi\Delta} \int_{-\infty}^{\infty} \frac{dq}{2\pi} q \sinh\left(\frac{q\pi}{2}\right) \frac{\left|\Gamma\left(\Delta + i\frac{q}{2} + 1\right)\right|^2 \left|\Gamma\left(\Delta + i\frac{q}{2}\right)\right|^4}{\Gamma(2\Delta + 2)\left(\Gamma(2\Delta)\right)^2}.$$
(67)

The expression above is real and non-positive for $n > 1$. Now, we are interested in the coefficient $C$ defined as

$$C \equiv \lim_{n\to 1} \frac{1}{n-1} \sum_{i<j<k} C_{ij}C_{jk}C_{ki} = \lim_{n\to 1} \frac{2^{6\Delta}}{3} R^{6\Delta} \frac{n\, C_n}{n-1},$$
(68)

where the second equality follows from (46). From (67) one finds

$$\frac{C}{R^{6\Delta}} = -\frac{2^{6\Delta}}{24\pi\Delta} \int_{-\infty}^{\infty} \frac{dq}{2\pi} q \sinh\left(\frac{q\pi}{2}\right) \frac{\left|\Gamma\left(\Delta + i\frac{q}{2} + 1\right)\right|^2 \left|\Gamma\left(\Delta + i\frac{q}{2}\right)\right|^4}{\Gamma(2\Delta + 2)\left(\Gamma(2\Delta)\right)^2}.$$
(69)

Finally, one can check this reduces to

$$\frac{C}{R^{6\Delta}} = -\frac{2^{6\Delta}\Gamma\left(\Delta + \frac{1}{2}\right)^3}{12\pi\Gamma\left(3\Delta + \frac{3}{2}\right)},$$
(70)

which leads to (33).

# 3 Mutual and tripartite information for fermions

In the previous section we derived a formula for the leading long distance contribution to the tripartite information for disjoint spheres (34). Such result was obtained for a generic CFT with a scalar as its lowest scaling dimension operator. However, in general, such an operator can have arbitrary spin. In this section we study how the above analysis gets modified when the lowest dimension operator is fermionic. This case is of special interest due to the fact that in two dimensions a free fermion has an identically vanishing tripartite information, $I_3 \equiv 0$, while a naive generalization to the formula (34) suggests a non-zero answer. This seems to be the case, due to the presence of a

---

[2] For a recent analysis of a similar analytic continuation see [27]. There is also an interesting analytic continuation in [28], where the authors continue a sum over three-point functions.

universal contribution coming from products of two-point functions —second term in (34). In this section we will show that such universal contribution vanishes identically for fermions. This fact will be later supported by a lattice analysis in Section 4.

Following Cardy [6], the twist operator $\tilde{\Sigma}_A$ is dominated at long distance by the product of two operators with the lowest scaling dimension in the theory. For spin half operators this implies

$$\tilde{\Sigma}_A \approx \sum_{j \neq i} C_{ij}^{A,\alpha\beta} \bar{\psi}_\alpha^i(r_A) \psi_\beta^j(r_A) \, , \tag{71}$$

where $i, j$ labels the sheets on which the spinor fields are located and $\alpha, \beta$ are spinor indices. The tensor structure for $C_{ij}^{A,\alpha\beta}$ was deduced in [9] to be

$$C_{ij}^{A,\alpha\beta} = a_{ij}^A \delta_{\alpha\beta} + b_{ij}^A n_A^\mu (\gamma_\mu)_{\alpha\beta} \, , \tag{72}$$

where $n_A^\mu$ is the vector normal to the spherical region. The authors of [9] further argued that $a_{ij}^A = 0$. In even dimensions this is the case due to chiral symmetry and in odd dimension this avoids parity violation. As in the scalar case, one can read off the undetermined coefficients $b_{ij}^A$, by studying the long distance behavior of the appropriate two point function in the presence of the twist operator. This is, one computes[3]

$$\bar{n}^\mu \text{Tr}[\langle \tilde{\Sigma}_A \bar{\psi}_\lambda^i(\bar{r}) (\gamma_\mu)_{\lambda\rho} \psi_\sigma^j(\bar{r}) \rangle] \quad \text{when} \quad |\bar{r} - r_A|^2 \to \infty \, . \tag{73}$$

In the above formula $\bar{r}$ is an arbitrary point chosen to be far away from $A$. Likewise, $\bar{n}$ is an arbitrary future directed time-like normal vector and thus one can chose it to have any particular direction as to simplify the above formula.

For the computation of (73) we need the two point function of the spinor fields

$$\langle \psi_\alpha(r_A) \bar{\psi}_\beta(r_B) \rangle = i(\gamma^\mu)_{\alpha\beta} \frac{(r_B - r_A)_\mu}{|r_B - r_A|^{2\Delta+1}} \, , \tag{74}$$

where $\Delta$ is the scaling dimension of the spinor field. We evaluate the quantity (73) by using (71) and (72) with $a_{ij}^A = 0$, which leads to

$$\bar{n}^\mu \langle \tilde{\Sigma}_A \bar{\psi}_\lambda^i(\bar{r}) (\gamma_\mu)_{\lambda\rho} \psi_\rho^j(\bar{r}) \rangle = \sum_{k \neq l} b_{kl}^A n_A^\nu (\gamma_\nu)_{\alpha\beta} \bar{n}^\mu (\gamma_\mu)_{\lambda\rho} \langle \bar{\psi}_\alpha^k(r_A) \psi_\beta^l(r_A) \bar{\psi}_\lambda^i(\bar{r}) \psi_\rho^j(\bar{r}) \rangle \, , \tag{75}$$

where the trace in (73) has been included implicitly. The four-point function factorizes into a product of two-point functions which can be evaluated using (74). The result is

$$
\begin{aligned}
\bar{n}^\mu \langle \tilde{\Sigma}_A \bar{\psi}_\lambda^i(\bar{r}) (\gamma_\mu)_{\lambda\rho} \psi_\rho^j(\bar{r}) \rangle &= -\frac{1}{r^{4\Delta}} b_{ji}^A n_A^\nu (\gamma_\nu)_{\alpha\beta} \bar{n}^\mu (\gamma_\mu)_{\lambda\rho} \hat{r}^\pi (\gamma_\pi)_{\rho\alpha} \hat{r}^\rho (\gamma_\sigma)_{\beta\lambda} \\
&= -\frac{1}{r^{4\Delta}} b_{ji}^A n_A^\nu \bar{n}^\mu \hat{r}^\pi \hat{r}^\sigma \text{Tr} (\gamma_\nu \gamma_\sigma \gamma_\mu \gamma_\pi) \\
&= -\left[ 2 (n_A \cdot \hat{r})(\bar{n} \cdot \hat{r}) - (n_A \cdot \bar{n}) \right] \frac{b_{ji}^A}{r^{4\Delta}} \, , 
\end{aligned}
\tag{76}
$$

---

[3]Notice that the correlator $\text{Tr}[\langle \tilde{\Sigma}_A \bar{\psi}_\rho^i(\bar{r}) \psi_\sigma^j(\bar{r}) \rangle]$ vanishes identically as it is proportional to the trace of a single gamma matrix

where we introduced the variables $r = |\bar{r} - r_A|$ and $\hat{r} = (\bar{r} - r_A)/|\bar{r} - r_A|$, and used the identity

$$\text{Tr}\left(\gamma_\alpha \gamma_\mu \gamma_\beta \gamma_\nu\right) = 2^{\left[\frac{d}{2}\right]}\left(\eta_{\alpha\mu}\eta_{\beta\nu} + \eta_{\alpha\nu}\eta_{\mu\beta} - \eta_{\alpha\beta}\eta_{\mu\nu}\right) \tag{77}$$

in the last line. Equation (76) can be inverted to obtain the coefficient $b_{ji}^A$ in terms of the correlator in (73) when $r$ goes to infinity as

$$b_{ji}^A = -\lim_{r\to\infty} 2^{-\left[\frac{d}{2}\right]} r^{4\Delta} \frac{\bar{n}^\mu \langle \tilde{\Sigma}_A \bar{\psi}_\lambda^i(\bar{r})\left(\gamma_\mu\right)_{\lambda\rho}\psi_\rho^j(\bar{r})\rangle}{\left[2\left(n_A \cdot \hat{r}\right)\left(\bar{n} \cdot \hat{r}\right) - \left(n_A \cdot \bar{n}\right)\right]}. \tag{78}$$

## 3.1 Mutual information

We can write down expressions for the leading term in the mutual information and tripartite information respectively in terms of the coefficients $b_{ij}$ from the twist operator expansion (71). We start with the mutual information $I(A, B)$ whose leading term, according to (7), (71) and (72) is given by

$$I(A, B) = \lim_{n\to 1} \frac{1}{n-1} \sum_{j\neq i}\sum_{l\neq k} b_{ij}^A b_{kl}^B n_A^\mu n_B^\nu \langle \bar{\psi}_\alpha^i(r_A)\left(\gamma_\mu\right)_{\alpha\beta}\psi_\beta^j(r_A)\bar{\psi}_\rho^k(r_B)\left(\gamma_\nu\right)_{\rho\sigma}\psi_\sigma^l(r_B)\rangle + \dots \tag{79}$$

Following the same steps used to obtain the coefficient $b_{ji}^A$ in (76), one can reduce the above expression to

$$I(A, B) = 2^{\left[\frac{d}{2}\right]+1} \frac{\left(2(n_A \cdot \hat{r})(n_B \cdot \hat{r}) - (n_A \cdot n_B)\right)}{r^{4\Delta}} \left(\lim_{n\to 1}\frac{1}{2(1-n)}\sum_{j\neq i} b_{ij}^A b_{ji}^B\right) + \dots, \tag{80}$$

where here $r = |r_B - r_A|$ and $\hat{r} = (r_B - r_A)/|r_B - r_A|$. With a bit of extra work it can be shown that the analytic continuation of the sum over $b_{ij}^A b_{ji}^B$ in the $n$ going to 1 limit (the last factor in (80)), equals the analogous coefficient for the scalar (18). Thus, the final long-distance result for the mutual information coincides with the one presented in [9] —including the tensor structure— as well as with the earlier work of [8]. This is

$$I(A, B) = 2^{\left[\frac{d}{2}\right]+1} \frac{\sqrt{\pi}}{4}\frac{\Gamma\left(2\Delta + 1\right)}{\Gamma\left(2\Delta + \frac{3}{2}\right)}\left[2(n_A \cdot \hat{r})(n_B \cdot \hat{r}) - (n_A \cdot n_B)\right]\frac{R_A^{2\Delta} R_B^{2\Delta}}{r^{4\Delta}} + \dots \tag{81}$$

## 3.2 Tripartite information

Now, we would like to study the analogous long-distance behavior of the tripartite information for conformal spinors. We start with the expression for the tripartite information given in (26). Using (71) and (72) we can write the leading term of (26) in terms of sums of six point functions as

$$I_3(A, B, C) \sim \lim_{n\to 1}\frac{1}{n-1}\sum_{j\neq i}\sum_{l\neq k}\sum_{m\neq n} b_{ij}^A b_{kl}^B b_{mn}^C n_A^\mu\left(\gamma_\mu\right)_{\alpha\beta} n_B^\nu\left(\gamma_\nu\right)_{\rho\sigma} n_C^\lambda\left(\gamma_\lambda\right)_{\pi\xi}$$
$$\times \langle \bar{\psi}_\alpha^i(r_A)\psi_\beta^j(r_A)\bar{\psi}_\rho^k(r_B)\psi_\sigma^l(r_B)\bar{\psi}_\pi^m(r_C)\psi_\xi^n(r_C)\rangle. \tag{82}$$

The above six-point function factorizes into products of two-point functions, there are no three-point function terms for spinors. Let us write down the factorization in question explicitly

$$\langle \bar{\psi}^i_\alpha(r_A)\psi^j_\beta(r_A)\bar{\psi}^k_\rho(r_B)\psi^l_\sigma(r_B)\bar{\psi}^m_\pi(r_C)\psi^n_\xi(r_C)\rangle$$

$$= -\delta^{il}\delta^{jm}\delta^{kn}\langle\bar{\psi}^i_\alpha(r_A)\psi^l_\sigma(r_B)\rangle\langle\psi^j_\beta(r_A)\bar{\psi}^m_\pi(r_C)\rangle\langle\bar{\psi}^k_\rho(r_B)\psi^n_\xi(r_C)\rangle$$

$$+\delta^{in}\delta^{jk}\delta^{lm}\langle\bar{\psi}^i_\alpha(r_A)\psi^n_\xi(r_C)\rangle\langle\psi^j_\beta(r_A)\bar{\psi}^k_\rho(r_B)\rangle\langle\psi^l_\sigma(r_B)\bar{\psi}^m_\pi(r_C)\rangle$$

$$= -\frac{i}{r^{2\Delta}_{AB}r^{2\Delta}_{AC}r^{2\Delta}_{BC}}\Big[ -\delta^{il}\delta^{jm}\delta^{kn}\hat{r}^\tau_{AB}\hat{r}^\eta_{AC}\hat{r}^\chi_{BC}(\gamma_\tau)_{\sigma\alpha}(\gamma_\eta)_{\beta\pi}(\gamma_\chi)_{\xi\rho}$$

$$+\delta^{in}\delta^{jk}\delta^{lm}\hat{r}^\tau_{AC}\hat{r}^\eta_{AB}\hat{r}^\chi_{BC}(\gamma_\tau)_{\xi\alpha}(\gamma_\eta)_{\beta\rho}(\gamma_\chi)_{\sigma\pi}\Big], \qquad (83)$$

where we used (74) in the second equality, $\hat{r}_{AB} \equiv (r_B - r_A)/|r_R - r_A|$, $r_{AB} \equiv |r_B - r_A|$, and similarly for the $AC$ and $BC$ combinations. Plugging (83) into (82), and after a bit of algebra we find

$$I_3(A,B,C) \sim \left(\lim_{n\to 1}\frac{1}{n-1}\sum_{i\neq j\neq k}b^A_{ij}b^B_{jk}b^C_{ki}\right)\frac{i}{r^{2\Delta}_{AB}r^{2\Delta}_{AC}r^{2\Delta}_{BC}}\hat{r}^\tau_{AB}\hat{r}^\eta_{AC}\hat{r}^\chi_{BC}n^\mu_A n^\nu_B n^\lambda_C$$

$$\left\{\text{Tr}\left[\gamma_\mu\gamma_\eta\gamma_\lambda\gamma_\chi\gamma_\nu\gamma_\tau\right] - \text{Tr}\left[\gamma_\tau\gamma_\nu\gamma_\chi\gamma_\lambda\gamma_\eta\gamma_\mu\right]\right\}.$$

$$(84)$$

We expect the analytically continued sum (the first term in brackets) to be related to the analogous coefficient for the scalar case (33). However, the term in curly brackets is identically equal to zero, and thus we conclude that the analogue contribution to the tripartite information obtained for scalars (25) identically vanishes for spinors

$$I_3(A,B,C) = 0 + \dots \qquad (85)$$

Therefore, the tripartite information at long distances must decay faster than $(R/r)^{6\Delta}$ when the lowest scaling dimension in the CFT is a spinor with scaling dimension $\Delta$. This is indeed the case for $2d$ free fermions as $I_3 \equiv 0$.[4] For free fermions in three dimensions we find via a lattice computation presented in Section 4, that

$$I_3 \sim (R/r)^{6\Delta_f+1} \qquad (86)$$

(where in that case $\Delta_f = 1$), which is consistent with the above result. We expect eq. (86) to be the leading-order scaling for theories with a fermion as their lowest-dimensional primary. An alternative possibility would involve an additional primary with a scaling dimension $\Delta_f < \tilde{\Delta} < \Delta_f + 1/6$ which would then give rise to a leading scaling $I_3 \sim (R/r)^{\tilde{\Delta}}$ instead. Observe that the difference in the leading power of the tripartite information between theories with a scalar or a fermion as their lowest-dimensional operator is somewhat different from the mutual information situation. In that case, the leading term is $\sim r^{-4\Delta}$ regardless of the spin of the lowest-dimensional primary —the only difference being an overall tensorial structure which changes as a function of the spin [9].

---

[4]For dimensions higher than two, free fermions are known not to be extensive [18]. Nevertheless, it is interesting to notice that free fermions are close to be so, as it can be seen from a comparison between the varios charges associated to the free fermion theory and the so called "Extensive Mutual Information model" [18, 19].

# 4 Lattice calculations in $(2+1)$ dimensions

In this section we perform some checks of our analytic results in the case of three-dimensional free fields. In particular, for the free scalar we verify that the long-distance scaling of the tripartite information is $I_3 \sim (R/r)^3$ and that the coefficient in the case of disk entangling regions matches our analytic prediction with reasonable precision. In the case of the fermion, we verify that the analogous long-distance scaling is $I_3 \sim (R/r)^7$, in agreement with our result that the naive leading scaling $I_3 \sim (R/r)^6$ does not hold due to the vanishing of the involved tensorial structures. The coefficient of the leading term for disk regions is also evaluated numerically for the free fermion.

## 4.1 Long-distance scaling for free scalars and fermions

Let us start with the free scalar. Consider a square lattice of $N$ points and a set of scalar fields and momenta $\phi_i, \pi_j$, $i, j = 1, \ldots, N$ satisfying canonical commutation relations, $[\phi_i, \pi_j] = i\delta_{ij}$, $[\phi_i, \phi_j] = [\pi_i, \pi_j] = 0$. Given a Gaussian state $\rho$, consider the two-point correlators $X_{ij} \equiv \mathrm{tr}(\rho\phi_i\phi_j)$, $P_{ij} \equiv \mathrm{tr}(\rho\pi_i\pi_j)$. Then, the entanglement entropy corresponding to a region $A$ can be obtained from the restrictions of $X_{ij}$ and $P_{ij}$ to the sites belonging to such a region as

$$S(A) = \mathrm{tr}\left[(C_A + 1/2)\log(C_A + 1/2) - (C_A - 1/2)\log(C_A - 1/2)\right], \tag{87}$$

where $C_A \equiv \sqrt{X_A P_A}$ and we denote $(X_A)_{ij} \equiv X_{ij}$, $(P_A)_{ij} = P_{ij}$ with $i, j \in A$.

Here we will work in $d = 2+1$, so each index $i$ corresponds to coordinates in a two-dimensional lattice. The free-scalar lattice Hamiltonian can be written as

$$H = \frac{1}{2} \sum_{n,m=-\infty}^{\infty} \left[\pi_{n,m}^2 + (\phi_{n+1,m} - \phi_{n,m})^2 + (\phi_{n,m+1} - \phi_{n,m})^2\right], \tag{88}$$

where we set the lattice spacing to one. Expressions for $X_{(x_1,y_1),(x_2,y_2)}$ and $P_{(x_1,y_1),(x_2,y_2)}$ for the vacuum state can be found in [29] and read

$$X_{(0,0),(i,j)} = \frac{1}{8\pi^2} \int_{-\pi}^{\pi} \mathrm{d}x \int_{-\pi}^{\pi} \mathrm{d}y \frac{\cos(ix)\cos(jy)}{\sqrt{2(1-\cos x) + 2(1-\cos y)}}, \tag{89}$$

$$P_{(0,0),(i,j)} = \frac{1}{8\pi^2} \int_{-\pi}^{\pi} \mathrm{d}x \int_{-\pi}^{\pi} \mathrm{d}y \cos(ix)\cos(jy)\sqrt{2(1-\cos x) + 2(1-\cos y)}. \tag{90}$$

Using these expressions, we can evaluate the tripartite information of lattice regions $A$, $B$, $C$ using eq. (87) and the general expression eq. (2).

The story is analogous for the free fermion. We start with fermionic fields $\psi_i$, $i = 1, \ldots, N$ defined at the lattice sites and satisfying canonical anticommutation relations, $\{\psi_i, \psi_j^\dagger\} = \delta_{ij}$. For a Gaussian density matrix $\rho$, we define the correlators matrix $D_{ij} \equiv \mathrm{tr}(\rho\psi_i\psi_j^\dagger)$. Then, the entanglement entropy for some region $A$ can be computed from the restriction of $D_{ij}$ to the corresponding lattice sites as

$$S(A) = -\mathrm{tr}\left[D_A \log D_A + (1 - D_A)\log(1 - D_A)\right]. \tag{91}$$

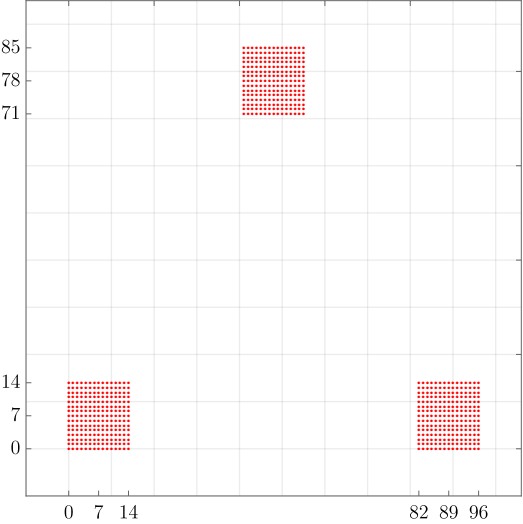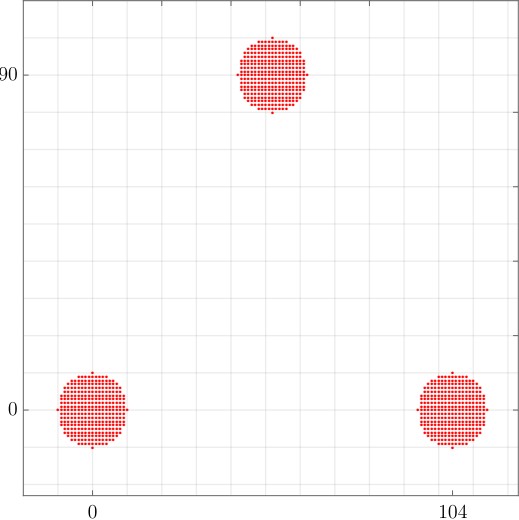

Figure 3: We show two examples of the equilateral-triangle lattice configurations considered. In the left, three squares of $15^2 = 225$ points separated a distance of $\simeq 82$ points. In the right, three disks of $317 \sim \pi 10^2$ points separated a distance of $\simeq 104$ points. The configurations are chosen so that the distances between each pair of centers are very similar. For instance, the distance between each of the lower squares and the upper one is $\sqrt{(82/2)^2 + 71^2} = 81.9878$. Similarly, the separation between each of the lower disks and the upper one is $\sqrt{(104/2)^2 + 90^2} = 103.942$.

The three-dimensional lattice Hamiltonian we consider for the free fermion reads

$$H = -\frac{i}{2} \sum_{n,m} \left[ \left( \psi_{m,n}^\dagger \gamma^0 \gamma^1 (\psi_{m+1,n} - \psi_{m,n}) + \psi_{m,n}^\dagger \gamma^0 \gamma^2 (\psi_{m,n+1} - \psi_{m,n}) \right) - h.c. \right] , \tag{92}$$

and the vacuum-state correlators read in this case

$$D_{(n,k),(j,l)} = \frac{1}{2} \delta_{n,j} \delta_{kl} - \int_{-\pi}^{\pi} dx \int_{-\pi}^{\pi} dy \frac{\sin(x)\gamma^0\gamma^1 + \sin(y)\gamma^0\gamma^2}{8\pi^2 \sqrt{\sin^2 x + \sin^2 y}} e^{i(x(n-j)+y(k-l))} . \tag{93}$$

In all cases, we restrict ourselves to configurations consisting of identical entangling regions which we separate forming approximate the vertices of equilateral triangles — see Fig. 3 for a couple of examples corresponding to square and disk regions.

Our first goal is to determine the power of the scaling of the tripartite information with the ratio $R/r$ for both theories. In order to do that, we consider square-shaped lattice regions of various side lengths $R$ and fix the distance $r$. Then, we plot the resulting data points against various possible powers of $(R/r)$. The idea is that whenever the right power is chosen, the points should follow a linear relation.

The results are plotted in Fig. 4. In the case of the scalar, we observe that a linear fit of the data points with respect to $(R/r)^3$ sits on top of the data points, whereas the $(R/r)^2$ and $(R/r)^4$ scalings are ruled out. In the case of the fermion, we observe that the naive $(R/r)^6$ scaling is disfavored by our numerical calculations, in agreement with our observation that this putative leading term is in fact absent. The next candidate leading power, $(R/r)^7$, is on the other hand the

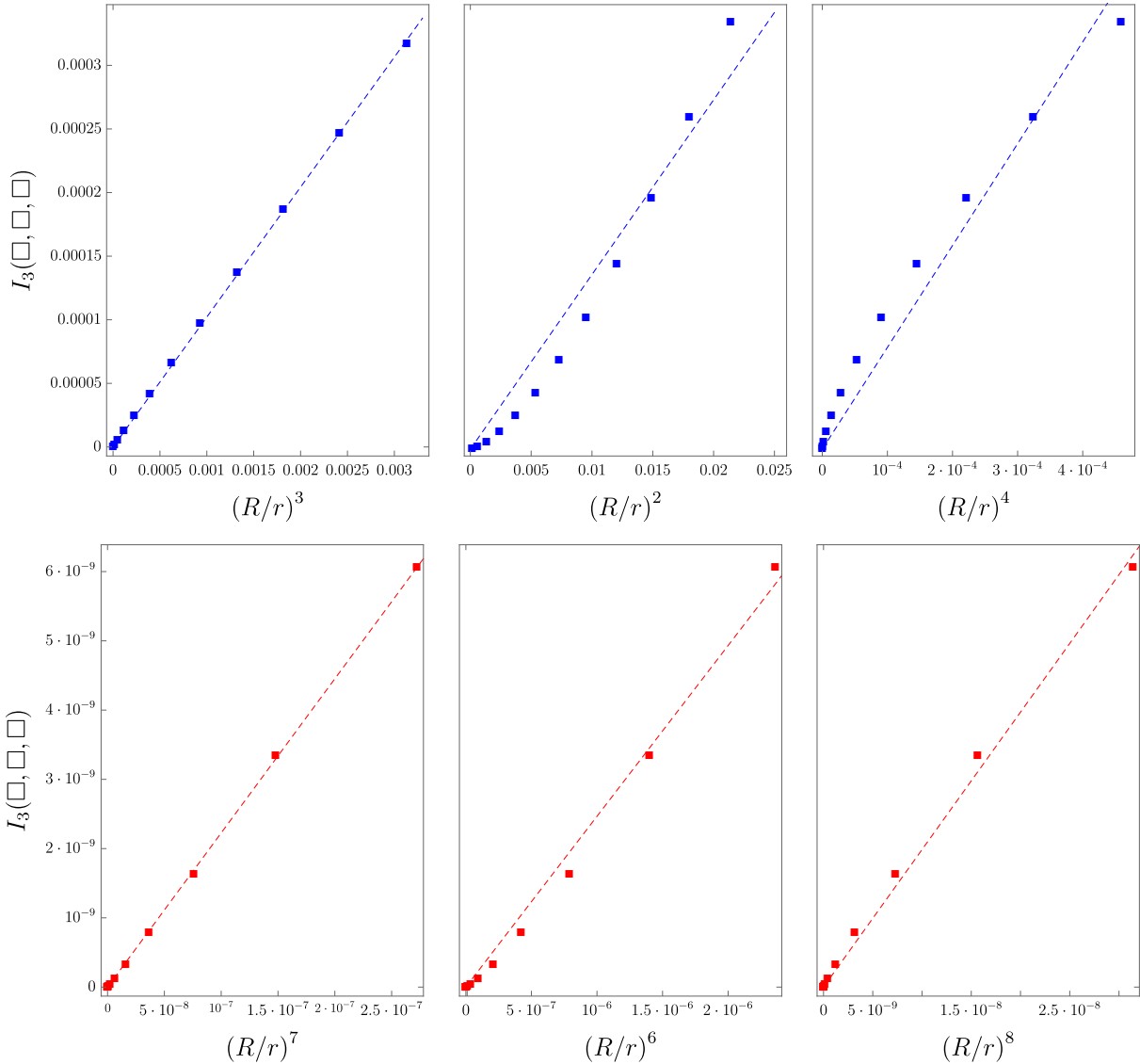

Figure 4: (Upper row) For a free scalar field, we plot $I_3(A, B, C)$ for three squares of equal size for several values of $(R/r)$ as a function of possible different powers of such ratio (data points). The $(R/r)^2$ and $(R/r)^4$ scalings are clearly off, whereas the $(R/r)^3$ one does a very good job in fitting the data linearly, as expected from our analytic computations. (Lower row) Same quantity for a free fermion. In this case, the differences between the possible scalings are not so neat, but it is nonetheless manifest that the $(R/r)^7$ linear fit is the best one.

winner of this comparison, strongly suggesting that in the case of the fermion, the long-distance behavior of the tripartite information is $I_3^{\text{ferm}} \propto (R/r)^7$. Observe also that both the scalar and the fermion have a tripartite information which is positive in the long-distance regime —namely, their mutual informations are non-monogamous.

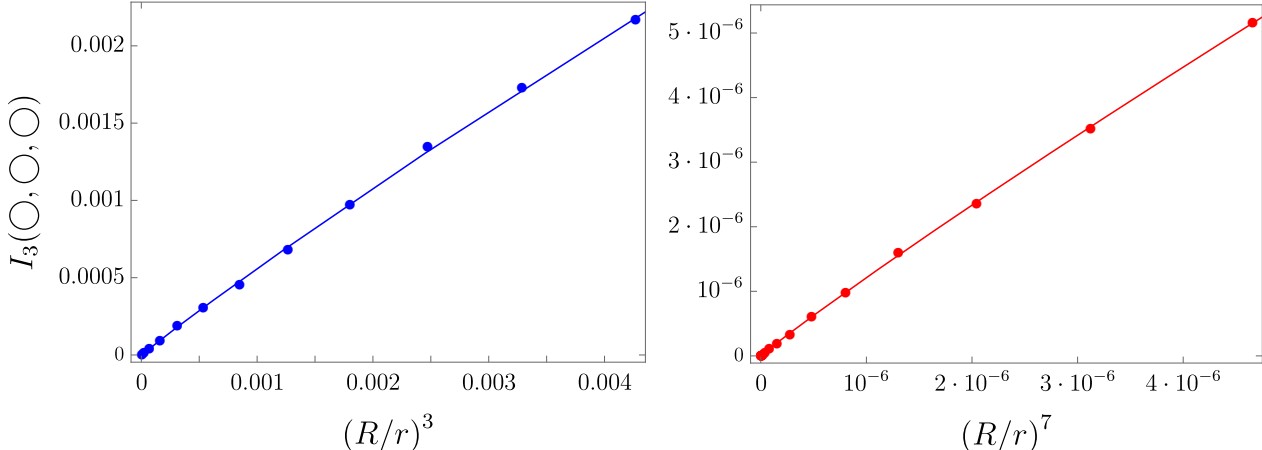

Figure 5: (Left) For a free scalar, we plot the tripartite information for three disks of radius $R$ positioned at the vertices of an equilateral triangle os side $r$ as a function of $(R/r)^3$. (Right) Same for a free fermion as a function of $(R/r)^7$. In both plots the solid lines correspond to fits which include a linear term plus a subleading correction as explained in the main text.

## 4.2 Three-disks coefficient

One of the results that follow from our analysis in the previous section is that the coefficient corresponding to the leading term in the long-distance expansion of the tripartite information in the case of three disks is $2/\pi$. Here we verify this prediction from a lattice calculation and perform the analogous analysis in the case of a free fermion.

In the left plot of Fig. 5 we show the results for various configurations of radius-$R$ disks positioned at the vertices of equilateral triangles of side $r$ as a function of $(R/r)^3$, which is the leading power in the long-distance regime, as we have learnt. At subleading order, we expect a contribution proportional to $(R/r)^4$, so in order to extract the coefficient of the leading term, we fit the data points to a function of the form $I_3 = \alpha_3 x + \alpha_4 x^{4/3}$ where $x \equiv (R/r)^3$. The resulting curve is shown in Fig. 5 and approximates all points rather well. The coefficients of the fit read, respectively, $\alpha_4 \simeq -0.741$ and

$$\alpha_3 \simeq 0.6325 = 0.9935 \cdot \frac{2}{\pi} \,, \tag{94}$$

which is an excellent agreement with the analytic result.

We repeat the analysis in the case of the free fermion. For that, we fit the data points to a function of the form $I_3 = \beta_7 x + \beta_8 x^{8/7}$, which assumes a subleading piece in the tripartite information scaling with $\sim (R/r)^8$. The fit is again excellent and appears in the right plot of Fig. 5. For the corresponding coefficients we find $\beta_8 \simeq -3.089$ and

$$\beta_7 \simeq 1.641 \,, \tag{95}$$

which —just like for the scalar— is a positive number and therefore corresponds to a non-monogamous mutual information (as anticipated in the case of the square regions). It would be interesting to

compute $\beta_7$ analytically and compare it with this numerical result.[5]

# 5 Discussion

In this paper we have shown how to compute the tripartite information in a CFT in an expansion for long distances of the involved regions. A more detailed summary of our main results can be found at the end of the introduction. We end with two comments. The first discusses these results as part of the program aiming at bootstrapping CFT data from entropy quantities. The second discusses what the results teach us about the monogamy condition in a CFT.

## 5.1 CFT data from mutual information

In [30] it was found that the mutual information for disjoint spherical regions in a CFT has an expansion in terms of conformal blocks of the form

$$I(A,B) = \sum_{\Delta,J} b_{\Delta,J} G_{\Delta,J}(u,v) \,, \tag{96}$$

where $\{\Delta, J\}$ is the set of replica primary operators which contribute to the Rényi mutual information and survives the $n \to 1$ limit. $G_{\Delta,J}(u,v)$ is the conformal block associated to the respective replica primary and it is written naturally in terms of the conformal ratios $u, v$. Also, $b_{\Delta,J}$ is a proportionality coefficient. The conformal ratios are constructed from the tips of the causal cones defining the spheres. For example, for a sphere $A$, $x_A^+$ denotes the future causal tip while $x_A^-$ denotes the past causal tip. The explicit expression is

$$u = \frac{|x_A^+ - x_A^-|^2 |x_B^+ - x_B^-|^2}{|x_B^- - x_A^-|^2 |x_B^+ - x_A^+|^2}, \quad v = \frac{|x_B^+ - x_A^-|^2 |x_A^+ - x_B^-|^2}{|x_B^- - x_A^-|^2 |x_B^+ - x_A^+|^2} \,. \tag{97}$$

Equation (96) comes from an OPE block expansion of the twist operator in the replica theory as reviewed in appendix A. Interestingly, knowledge of the mutual information for disjoint spheres can be used to "bootstrap" part of the operator content in the replica theory.[6] Such procedure was outlined in [19] where we used it to rule out the "Extensive Mutual Information model" as corresponding to a CFT in $d \geq 3$.

The "bootstrapping" procedure is the following. We consider the long-distance limit of each conformal block, in the usual cross-ratio variables $u, v$. This corresponds to the $u \to 0$ and $v \to 1$ limits, which in terms of the physical parameters is [9]

$$u \sim \frac{16 R_A^2 R_B^2}{L^4}, \quad v \sim 1 - \frac{8 R_A R_B}{L^2} \left[ 2 \left( n_A \cdot \hat{r} \right) \left( n_B \cdot \hat{r} \right) - n_A \cdot n_B \right] \,. \tag{98}$$

---

[5]We point out that a function of the form $I_3 = \tilde{\beta}_7 x + \tilde{\beta}_9 x^{9/7}$ produces an almost identical fit for coefficients $\tilde{\beta}_7 \simeq 1.399$ and $\tilde{\beta}_9 \simeq -9.796$. Given that the naive $\mathcal{O}(R/r)^6$ term is actually absent for the fermion, it does not seem impossible that the $\mathcal{O}(R/r)^8$ term does not appear either. In that case, the exact coefficient for the leading piece would be closer to $\tilde{\beta}_7$ rather than to $\beta_7$.

[6]This formula does not necessarily include all the primary operators that appear in the replica theory as there might be many operators which do not contribute to the mutual information. However, the replica primaries which are simply related to the primary operators of the seeding CFT will always appear in the mutual information. For instance, operators of the form $\mathcal{O}_i \mathcal{O}_j$ with $i, j$ replica indices always appear in the mutual information.

In that case

$$\lim_{u\to 0, v\to 1} G_{\Delta, J}(u, v) \sim c_{d,J} u^{\frac{\Delta}{2}} C_J^{\frac{d}{2}-1} \left[ \frac{v-1}{2u^{1/2}} \right]$$

$$= c_{d,J} \left( \frac{4R_A R_B}{L^2} \right)^{\Delta} C_J^{\frac{d}{2}-1} \left[ 2 (n_A \cdot \hat{r})(n_B \cdot \hat{r}) - n_A \cdot n_B \right], \qquad (99)$$

where the $C_J^{\frac{d}{2}-1}[x]$ are the Gegenbauer polynomials. Therefore, from (96) we see that the long-distance limit of $I(A,B)$ would be given by the long-distance limit of the leading conformal block, namely, the RHS of (99) for the smallest possible $\Delta$. Thus, from this term we can read off the corresponding scaling dimension $\Delta$ and spin $J$ of the smallest replica primary which contributes to the mutual information.

Next, we can subtract off the full leading conformal block appearing on the RHS of (96) from $I(A,B)$, which results in:

$$I^{(1)}(A,B) \equiv I(A,B) - b_{\Delta_1, J_1} G_{\Delta_1, J_1}(u,v) = \sum_{\Delta \neq \Delta_1, J \neq J_1} b_{\Delta, J} G_{\Delta, J}(u, v), \qquad (100)$$

where the super-index (1) in $I^{(1)}(A,B)$ indicates that we removed the first leading conformal block to the mutual information. After that, we can apply the described algorithm to $I^{(1)}(A,B)$, finding in this way the subleading replica primary operator that contributes to $I(A,B)$. Possible degeneracies could also be accounted for by identifying the linear combination of Gegenbauer polynomials contributing to that order (which is possible by the completeness of the Gegenbauer polynomials). Some of the coefficients $c_{d,J}$ appearing in (99) can be obtained via an explicit computation using the framework developed in [9].

In summary, applying the above procedure one could reconstruct the set of primary replica operators that contributes to the mutual information, including their corresponding scaling dimensions $\Delta$'s and spins $J$'s. Via a detailed analysis of the possible replica operators that can be constructed from the original or seed CFT, one could invert the above data to obtain the set of primary operators, their scaling dimensions $\bar{\Delta}$'s and associated spins $\bar{J}$'s as well as possibly some of the OPE coefficients[7] $C_{ijk}$ of the seed CFT. Let us elaborate a bit further on that possibility.

Schematically, the replica primary operators can be constructed from the seed primaries in varios different ways. For example, some of them can include products of two seed primaries in different replicas with arbitrary number of derivatives in between

$$A^{\mu_1 \cdots \mu_n} \mathcal{O}_i \partial_{\mu_1} \cdots \partial_{\mu_n} \mathcal{O}_j, \qquad (101)$$

where the tensor structures $A^{\mu_1 \cdots \mu_n}$ may have different symmetries and $\mathcal{O}_i$ are scalars. For fermions there are also tensor structures one can be build from two-seed primary fermions, and which have non zero coefficients,

$$\bar{\psi}_i \gamma_\mu \psi_j, \quad \bar{\psi}_i \gamma_{\mu_1} \cdots \partial_{\mu_n} \psi_j, \qquad \cdots \qquad (102)$$

None of these replica primaries would have information about the structure coefficients $C_{ijk}$, and their coefficients depend only on the two-point function. However, there are replica primaries

---

[7]In this section we use $\bar{\Delta}$'s to represent the conformal dimensions of the seed theory operators while $\Delta$'s to represent the conformal dimensions of the replica theory.

formed by fields in more than two copies consistent with conservation laws and super-selection constraints, for example,

$$\mathcal{O}_i \mathcal{O}_j \mathcal{O}_k, \qquad \Psi_i \bar{\Psi}_j \Psi_k \bar{\Psi}_l \qquad \mathcal{O}_i \bar{\psi}_j \gamma_\mu \psi_k \,. \tag{103}$$

The contributions of these replica primaries would contain information about the OPE coefficients $C_{ijk}$ and thus the above procedure could in principle allow us to extract such CFT data from $I(A, B)$. Unfortunately, the procedure for these operators as well as replica primaries involving higher number of replica operators is significantly harder to use in practice than the ones that involve only two replicas. Therefore, one might deem this procedure unpractical for the purpose of obtaining the OPE coefficients.

Interestingly, our current work presents a complementary avenue for extracting the OPE coefficients. As opposed to what happens for the mutual information of two disjoint spheres, the tripartite information for three spheres at long distances receives contributions at the leading order from the replica primaries involving three replicas. For this reason, even the leading expression for the tripartite information includes also information about the OPE coefficients of the seed CFT as is manifest in (34). Thus, the reconstruction procedure derived from (96) can be complemented with an analogous one from (34) properly generalized to include all replica primaries, to facilitate the extraction of the full information of the seed CFT.

## 5.2   Monogamy condition and holography

From the expression of the long-distance tripartite information (34) it is easy to read off a condition for having monogamy of mutual information, $I_3 \leq 0$. In this geometric setup and in this regime the condition reduces to

$$(C_{\mathcal{O}\mathcal{O}\mathcal{O}})^2 \geq \frac{2^{6\Delta+1}\Gamma\left(\Delta + \frac{1}{2}\right)^3}{\pi^{3/2}\Gamma(3\Delta+1)} = \frac{2}{\Gamma\left(3\Delta+1\right)}\left(\frac{2\,\Gamma\left(2\Delta\right)}{\Gamma\left(\Delta\right)}\right)^3 \,. \tag{104}$$

The RHS of (104) is a growing function of $\Delta$ and its limiting value when $\Delta \to 0$ is 2. The asymptotic behavior for large $\Delta$ can be determined from the Stirling approximation, which gives

$$\frac{2}{\Gamma\left(3\Delta+1\right)}\left(\frac{2\,\Gamma\left(2\Delta\right)}{\Gamma\left(\Delta\right)}\right)^3 \sim \frac{4}{(3\pi\Delta)^{1/2}}\left(\frac{4}{3}\right)^{3\Delta} \,. \tag{105}$$

This approximation is an strict upper bound on the RHS of (104) and thus it is a good estimate on how large should $(C_{\mathcal{O}\mathcal{O}\mathcal{O}})^2$ be for the theory to be monogamous at large separations. This is a strong condition over $(C_{\mathcal{O}\mathcal{O}\mathcal{O}})^2$, which suggest that generically in QFT the mutual information for separated regions tends to be non-monogamous (if dominated by scalars[8]). Definitely, monogamous behavior could only hold far from a perturbative regime. This statement is in line with the observation that in a perturbative scheme the tripartite information is generically non-monogamous [31], although note that the latter statement was made in the context of entanglement in momentum space.

As mentioned in the introduction, for holographic theories the tripartite information is known to be monogamous at leading order in the large-$N$ parameter. However, the existence of RT phase

---

[8]But possibly also for fermions in view of our results in the rest of the paper.

transitions for disjoint regions implies that in our regime of interest —large separation— the RT contribution to the holographic tripartite information vanishes and thus its behavior is determined by the subleading contribution, which is given by the tripartite information of the associated dual bulk homology regions. Depending on the dual bulk theory, then, this tripartite information might be positive or negative, which renders the boundary mutual information to be generically non-monogamous. However, there is an interesting possibility, namely, one in which the bulk theory is itself holographic. These situations are known as *double holographic* [32, 33], and they have been the focus of important recent activity due to their relevance in the partial resolution to the black hole information paradox [34, 35]. In this context, one could imagine situations in which the bulk mutual information of the first dual theory is non vanishing at leading order, thus the RT surface of the second dual theory would be in the connected phase and therefore it would be necessarily monogamous (by the properties of the RT formula for the second holographic theory). In other words, in such geometric configurations the first bulk theory would be monogamous, and likewise would be its associated boundary theory. Indeed, boundary monogamy has been recently proved to hold at all orders in the large-$N$ expansion provided the bulk theory is also monogamous [36].[9] Unfortunately, in the strict large separation regime both first and second bulk RT surfaces would be in the disconnected phase and thus monogamy would not be guarantee even in double holography.

# Acknowledgements

It is a pleasure to thank Gonzalo Torroba for useful discussions. C.A. is specially grateful to Tomonori Ugajin for various discussions regarding tripartite information in CFT. This material is based upon work supported by the Simons Foundation through *It from Qubit: Simons Collaboration on Quantum Fields, Gravity, and Information*. H.C. acknowledges support from the National University of Cuyo, CNEA, and CONICET, Argentina.

# A    OPE block expansion of the tripartite information

In this appendix we want to comment on how to improve our result for the leading term of the tripartite information by including all descendent operators of the leading ones. As explained earlier, (26) represents the leading Rényi tripartite information as a correlator of twist operators. In that expression we have the following expansion for the non-local twist operators (10)

$$\tilde{\Sigma}_A^{(n)} = \sum_{\{k_j\} \neq \mathbb{I}} C_{\{k_j\}}^A \prod_{j=0}^{n-1} \Phi_{k_j}^{(j)}(r_A) \,. \tag{106}$$

However, one can improve the above ansatz by taking each primary operator $\prod_{j=0}^{n-1} \Phi_{k_j}^{(j)}(r_A)$ in the replica theory and adding all its descendants, in other words, by considering instead its associated OPE block.

The OPE block appears in the contribution of a primary operator to the OPE of two primaries in a general CFT. For example, when the primaries in question are scalars, say $\mathcal{O}_i(x)$ and $\mathcal{O}_j(0)$,

---

[9]See also [37] for a weaker statement proved in the context of quantum bit threads. Namely, holographic entropy cone inequalities in the bulk imply boundary monogamy.

with conformal dimensions $\Delta_i$, $\Delta_j$, then one can replace its product inside the expectation value of an arbitrary product of local operators by the following expansion

$$\mathcal{O}_i(x)\mathcal{O}_j(0) = \sum_k C_{ijk}|x|^{\Delta_k - \Delta_i - \Delta_j}\left(1 + b_1 x^\mu \partial_\mu + b_2 x^\mu x^\nu \partial_\mu \partial_\nu + \cdots\right)\mathcal{O}_k(0), \qquad (107)$$

provided all other operator insertions are located sufficiently apart from points $x$ and $0$.[10] The coefficients $b_n$ become independent of the conformal dimensions $\Delta_i$, $\Delta_j$ when $\Delta_i = \Delta_j$. In that case, the total contribution associated to a given $k$ will depend only on the conformal symmetry and the generating operator $\mathcal{O}_k$. Such contribution is known as the OPE block associated to $\mathcal{O}_k$,

$$\mathcal{B}_k(x,0) = |x|^{\Delta_k}\left(1 + b_1 x^\mu \partial_\mu + b_2 x^\mu x^\nu \partial_\mu \partial_\nu + \cdots\right)\mathcal{O}_k(0). \qquad (108)$$

There is a useful integral expression for this operator in cases in which the points $\{x, 0\}$, hereafter $\{x_1, x_2\}$ are time-like separated and therefore define a causal cone $D(x_1, x_2)$ with $\{x_1, x_2\}$ as its tips [39],

$$\mathcal{B}_k(x_1, x_2) = c_k \int_{D(x_1,x_2)} \mathrm{d}^d\xi \left(\frac{|x_1 - \xi||x_2 - \xi|}{|x_1 - x_2|}\right)^{\Delta_k - d}\mathcal{O}_k(\xi). \qquad (109)$$

There exists an analogous formula for the OPE block of an arbitrary primary operator in a symmetric spin $J$ representation $\mathcal{O}_{\mu_1 \cdots \mu_J}$, namely [40],

$$\mathcal{B}_{k,J}(x_1, x_2) = \frac{c_k}{(2\pi)^{\Delta_k - d}} \int_{D(x_1,x_2)} \mathrm{d}^d\xi \, |K|^{\Delta_k - d - J} K^{\mu_1} \cdots K^{\mu_J}\mathcal{O}_{k,\mu_1\cdots\mu_J}(\xi). \qquad (110)$$

Here, $K^\mu$ is the conformal killing vector that keeps the boundary of the causal cone fixed, and it is given by

$$K^\mu \partial_\mu = -\frac{2\pi}{(x_1 - x_2)^2}\left[(x_2 - \xi)^2(x_1^\mu - \xi^\mu) - (x_1 - \xi)^2(x_2^\mu - \xi^\mu)\right]\partial_\mu, \qquad (111)$$

and

$$|K| = 2\pi \frac{|x_1 - \xi||x_2 - \xi|}{|x_1 - x_2|}. \qquad (112)$$

The above is precisely the proposal of Long [30]. In short, the idea is to improve upon the expansion of (106) developed by Cardy by considering instead a basis of non-local operators associated to the entangling region. For a sphere $\mathbb{S}_A$, such operators are precisely the OPE blocks $(2R_A)^{-\Delta_k}\mathcal{B}_{k,J}(x_A^+, x_A^-)$ with $\{x_A^+, x_A^-\}$ as the tips of the causal development of the associated spherical region. The expansion would have the form

$$\tilde{\Sigma}_A^{(n)} = \sum_{\{k_j\}\neq\mathbb{I}} C_{\{k_j\}}^A \prod_{j=0}^{n-1} \frac{c_{k_j}(2R_A)^{-\Delta_{k_j}}}{(2\pi)^{\Delta_{k_j} - d}} \int_{D(x_A^+, x_A^-)} \mathrm{d}^d\xi \, |K|^{\Delta_{k_j} - d - J} K^{\mu_1} \cdots K^{\mu_J}\Phi_{k_j,\mu_1\cdots\mu_J}^{(j)}(\xi). \qquad (113)$$

The leading contributing primary operator $\Phi_{k_j,\mu_1\cdots\mu_J}^{(j)}(\xi)$ corresponds to the product of two primary operators associated to different sheets with the lowest scaling conformal dimension $\Delta$, this is,

$$\sum_{\{k_j\}\neq\mathbb{I}} C_{\{k_j\}}^A \prod_{j=0}^{n-1} \Phi_{k_j,\mu_1\cdots\mu_J}^{(j)}(\xi) = \sum_{ij}(2R_A)^{-2\Delta}C_{ij}\,\mathcal{O}^i(\xi)\mathcal{O}^j(\xi), \qquad (114)$$

---

[10]There is a technically precise sense in which the above replacement is accurate for observables with support outside the radius of convergence of the OPE [38].

and therefore, the leading contribution of (113) would be

$$\tilde{\Sigma}_A^{(n)} = \sum_{ij} C_{ij} \frac{c_{2\Delta}(2R_A)^{-2\Delta}}{(2\pi)^{2\Delta-d}} \int_{D_A} \mathrm{d}^d\xi \, |K|^{2\Delta-d} \mathcal{O}^i(\xi)\mathcal{O}^j(\xi) \,, \tag{115}$$

where we simplified our notation by defining $D_A \equiv D(x_A^+, x_A^-)$. The normalization constant $c_{2\Delta}$ satisfies

$$\frac{c_{2\Delta}(2R_A)^{-2\Delta}}{(2\pi)^{2\Delta-d}} \int_{D_A} \mathrm{d}^d\xi \, |K|^{2\Delta-d} = 1 \,. \tag{116}$$

If one replaces (115) into the formula for the mutual information as given in (14) one gets

$$I(A,B) = \frac{\sqrt{\pi}}{4} \frac{\Gamma\left(2\Delta+1\right)}{\Gamma\left(2\Delta+\frac{3}{2}\right)} \frac{c_{2\Delta}^2}{(2\pi)^{2(2\Delta-d)}} \int_{D_A} \mathrm{d}^d\xi_A \int_{D_B} \mathrm{d}^d\xi_B \frac{|K_A|^{2\Delta-d}|K_B|^{2\Delta-d}}{|\xi_A-\xi_B|^{4\Delta}} \,. \tag{117}$$

The double integral above can be identified with the conformal block associated to an intermediate scalar operator of dimension $2\Delta$ via $G_{\Delta_k,0}^d(u,v) = \langle \mathcal{B}_{\Delta_k}(x_A^+, x_A^-)\mathcal{B}_{\Delta_k}(x_B^+, x_B^-)\rangle$ which is a known relation between the conformal and OPE blocks, consistent with our normalizations. The above expression reduces to

$$I(A,B) = \frac{\sqrt{\pi}}{2^{4\Delta+2}} \frac{\Gamma\left(2\Delta+1\right)}{\Gamma\left(2\Delta+\frac{3}{2}\right)} G_{2\Delta,0}^d(u,v) \,, \tag{118}$$

which is the leading therm in the conformal block expansion of the mutual information [8]. In the above expressions $u$ and $v$ are the usual conformal ratios defined explicitly in (97). Adding all other possible replica primaries in the expansion of $\tilde{\Sigma}_A^{(n)}$ leads to the full conformal block expansion of the mutual information as described in Section 5.1 in the form of (96).

Similarly, we can replace (115) into the formula for the tripartite mutual information as given in (26), follow through all the analysis of Section 2 until the derivation of the analogous formula to (34) which in our present case is

$$I_3(A,B,C) = -\left[\frac{\sqrt{\pi}}{4} \frac{\Gamma(3\Delta+1)}{\Gamma\left(3\Delta+\frac{3}{2}\right)} \left(C_{\mathcal{OOO}}\right)^2 - \frac{2^{6\Delta}\Gamma\left(\Delta+\frac{1}{2}\right)^3}{2\pi\Gamma\left(3\Delta+\frac{3}{2}\right)}\right] \tag{119}$$

$$\times \int_{D_A}\int_{D_B}\int_{D_C} \frac{F_3(\xi_A,\xi_B,\xi_C)}{|\xi_A-\xi_B|^{2\Delta}|\xi_B-\xi_C|^{2\Delta}|\xi_A-\xi_C|^{2\Delta}} \mathrm{d}^d\xi_A\mathrm{d}^d\xi_B\,\mathrm{d}^d\xi_C$$

where we have introduced the function

$$F_3(\xi_A,\xi_B,\xi_C) \equiv \frac{c_{2\Delta}^3}{(2\pi)^{3(2\Delta-d)}} |K_A|^{2\Delta-d}|K_B|^{2\Delta-d}\,|K_C|^{2\Delta-d} \,. \tag{120}$$

This is our final formula for the long-distance tripartite information which includes the contribution of the leading OPE block in the twist operator expansion.

It would be interesting to explore whether the triple integral expression in (119) can be identified with an interesting object in the CFT as it happens to the analogous formula for the mutual information (118). Similarly, it would be interesting to study the full expansion of the tripartite information which includes all replica primaries that can contribute to the twist operator expansion in (113).

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
