# Peer review of "Tripartite information at long distances"

_SciPost Physics_

## Round 1 · Referee Report · Anonymous (Referee 1) · 2022-1-4

Strengths
1- considers an important and timely problem: the tripartite information in CFTs, holographic and otherwise 2- presents detailed calculations and many novel results, including the qualitatively novel fact that the leading contribution to the tripartite information depends on a certain OPE coefficient (and not just the spectrum) 3- includes a thorough analysis of the result, and a check against lattice calculations 4- is very clearly written and organized
Weaknesses
Report
Requested changes
Equation (22) has a typo in the form of a spurious + sign, which is somewhat confusing.

---

## Round 1 · Referee Report · Anonymous (Referee 2) · 2022-2-26

Strengths
1- interesting topic 2- analytic results and corresponding numerical checks
Weaknesses
Report
In particular, they focus on the asymptotic regime where the distances between the spheres are much larger than their radius.
The analysis is performed by extending to this interesting case the techniques discussed in refs. [6] and [7].
The main analytic result is eq. (34), as clearly stated also by the authors, and I find it insightful also the analysis involving the conformal spinors in sec. 2.3. It is also important the numerical analysis reported in sec. 4.2: it has allowed to check the analytic result corresponding to the massless scalar and to find a numerical prediction for the fermionic model.
I find this paper very interesting and well written; hence I strongly recommend its publication in Scipost.
Requested changes
I strongly suggest a little effort to improve the comparison of the results presented in this paper with the existing ones in 1+1 dimensions, starting e.g. from the following missing references:
1) the relevant paper 1011.5482, where the expansion of the mutual information in 1+1 CFT has been first studied and the method employed through the paper has been established, finding also the result (19) in one space dimension.
2) a qualitative comparison can be discussed between the results obtained in this paper with the ones for the tripartite information obtained in 1+1 dimensions e.g. in 1309.2189 and 1501.04311

---

## Editorial Decision

unknown